# Different viral effectors suppress hormone-mediated antiviral immunity of rice coordinated by OsNPR1

Hehong Zhang[1], Fengmin Wang[1], Weiqi Song[1], Zihang Yang[1], Lulu Li[1], Qiang Ma[1], Xiaoxiang Tan[1], Zhongyan Wei[1], Yanjun Li[1], Junmin Li[1], Fei Yan[1], Jianping Chen ®[1,2] ✉ & Zongtao Sun ®[1,2] ✉

Salicylic acid (SA) and jasmonic acid (JA) are plant hormones that typically act antagonistically in dicotyledonous plants and SA and JA signaling is often manipulated by pathogens. However, in monocotyledonous plants, the detailed SA-JA interplay in response to pathogen invasion remains elusive. Here, we show that different types of viral pathogen can disrupt synergistic antiviral immunity mediated by SA and JA via OsNPR1 in the monocot rice. The P2 protein of rice stripe virus, a negative-stranded RNA virus in the genus *Tenuivirus*, promotes OsNPR1 degradation by enhancing the association of OsNPR1 and OsCUL3a. OsNPR1 activates JA signaling by disrupting the OsJAZ-OsMYC complex and boosting the transcriptional activation activity of OsMYC2 to cooperatively modulate rice antiviral immunity. Unrelated viral proteins from different rice viruses also interfere with the OsNPR1-mediated SA-JA interplay to facilitate viral pathogenicity, suggesting that this may be a more general strategy in monocot plants. Overall, our findings highlight that distinct viral proteins convergently obstruct JA-SA crosstalk to facilitate viral infection in monocot rice.

Plants are frequently attacked by a variety of viruses, which seriously restrict plant physiological processes and cause abnormal host plant development[1–3]. Rice is vulnerable to infection by taxonomically diverse RNA viruses that cause enormous losses in crop yield and quality. One of the most destructive is rice stripe virus (RSV), a negative-strand RNA virus classified in the genus *Tenuivirus* (family *Phenuiviridae*)[4]. RSV is transmitted by *Laodelphax striatellus* (small brown planthopper, SBPH) and has a single-stranded RNA (ssRNA) genome of four segments encoding seven proteins by an ambisense expression strategy. Another important virus, Southern rice black-streaked dwarf virus (SRBSDV), belongs to the genus *Fijivirus* (family *Reoviridae*). It has 10 segments of double-stranded RNA and is transmitted by *Sogatella furcifera* (white-backed planthopper, WBPH)[5]. Rice stripe mosaic virus (RSMV) has emerged more recently and is not closely related to either RSV or SRBSDV. It has an undivided negative-sense ssRNA genome, is classified in the genus *Cytorhabdovirus* (family *Rhabdoviridae*) and is transmitted by the leafhopper *Recilia dorsalis*[6]. These viruses are widespread in rice growing area causing severely abnormal plant growth (necrotic stripes, leaf wilting and dwarfism) and inflicting serious damage on rice production[4,7]. We recently reported that these unrelated rice viruses share a common pathogenic strategy employing a class of independently evolved viral effectors to manipulate the key components of plant hormone signaling pathways in ways that facilitated infection by their respective viruses[8–11]. These unrelated viral effector proteins were identified as RSV P2, SRBSDV SP8, and RSMV M and they all interacted with the same targets, namely auxin response transcription factor OsARF17, JA signaling central components OsJAZ and OsMYC2/3, and GA signaling key component SLR1[8–10].

[1]State Key Laboratory for Managing Biotic and Chemical Threats to the Quality and Safety of Agro-products, Key Laboratory of Biotechnology in Plant Protection of MARA and Zhejiang Province, Institute of Plant Virology, Ningbo University, Ningbo 315211, China. [2]These authors contributed equally: Jianping Chen, Zongtao Sun. ✉e-mail: jianpingchen@nbu.edu.cn; sunzongtao@nbu.edu.cn

During the co-evolutionary arms race between virus and host, plants have developed a sophisticated and multifaceted immune system[12–14] that includes a complex interplay between different phytohormones, particularly jasmonate (JA) and salicylate (SA) signaling. The mutually antagonistic interlinking of the SA and JA pathways has been well documented in the model plant *Arabidopsis*[15,16]. However, these two pathways are not always antagonistic. In *Arabidopsis*, pathogen-induced SA and SA receptors NPR3/4 activate the JA signaling pathway and then promote effector-triggered immunity-associated programmed cell death[17]. In rice, SA and JA signaling both confer resistance to biotrophic pathogens[18]. In response to viral pathogens, the SA and JA pathways are widely reported to play essential roles in plant antiviral defense[19–22]. Our previous research showed that JA signaling cooperated with brassinosteroids (BR), abscisic acid (ABA), and auxin pathways to activate rice antiviral immunity[23–25]. A recent study also demonstrated that JA signaling and RNA silencing synergistically enhance antiviral defense in rice[26]. Similarly, SA was reported to enhance plant defense against rice viruses. The *hypersensitive-induced reaction gene* (*HIR3*) acts to regulate plant resistance against RSV infection via an SA-dependent pathway[27]. A sulfotransferase *STV11*, which catalyzes the conversion of SA to sulfonated SA (SSA), confers durable resistance to RSV[28]. These findings suggest that the interplay between SA and JA may be positive or negative depending on the particular pathogen-host combination. However, although there is a wealth of information on the antagonistic interaction, the detailed mechanism of SA-JA synergism is obscure, especially in monocot plants.

NONEXPRESSER OF PR GENES1 (NPR1) is the master regulator in SA signaling and has been well-characterized in *Arabidopsis*. Recent research revealed that NPR1 forms a bird-shaped homodimer, including an N-terminal Broad-complex, Tramtrack and Bric-à-brac (BTB) domain, a BTB and carboxyterminal Kelch helix bundle, four ankyrin repeats (ANK) and a C-terminal disordered SA-binding domain[29]. In normal cells, NPR1 predominantly exists as an oligomeric complex in the cytoplasm. Upon pathogen infection, increased SA levels trigger a change in cellular reduction potential and NPR1 is reduced from an oligomer to a monomer[30]. Monomeric NPR1 is translocated to the nucleus to activate gene transcription as a coactivator[31]. Because NPR family proteins have no DNA binding domains, NPR1 must interact with the transcription factors TGA (TGACG SEQUENCE-SPECIFIC BINDING PROTEIN) to induce the expression of *PR* genes[32]. Although many studies have shown that NPR1 positively regulates plant immunity and protects plants against diseases[33–36], there are relatively few studies of bacterial or fungal effectors directly manipulating NPR1-mediated plant defense[37,38]. It is currently not known whether other microbial pathogens, such as viruses, modulate NPR1 to benefit infection.

In this study, we found that rice stripe virus P2 protein directly interacts with OsNPR1 and interferes with its oligomerization. P2 promotes OsNPR1 degradation by enhancing the association of OsNPR1 and the cullin-RING ubiquitin ligases OsCUL3a in a SA-independent manner. OsNPR1 activated JA signaling by physically associating with the OsMYC2-JAZ complex to cooperatively modulate rice antiviral immunity. Intriguingly, we demonstrate that distinct viral proteins of unrelated rice viruses interfere with OsNPR1-mediated SA-JA interplay to facilitate viral pathogenicity. Together, our findings provide insight into the crosstalk mechanism between SA and JA signaling and deepen our understanding of how different viral pathogens commonly manipulate hormone-mediated antiviral immunity.

## Results

### RSV P2 protein interacts with OsNPR1 in a SA-independent manner

We have previously shown that RSV P2 directly interferes with auxin and JA signaling-mediated antiviral defense to benefit viral infection[8,9]. To investigate whether other key host factors are targeted by the P2

protein, we used it as a bait to screen a rice cDNA library in a yeast two-hybrid (Y2H) assay. Preliminary results showed that OsNPR1 interacted with the P2 protein (Fig. 1a). Coimmunoprecipitation (Co-IP) and bimolecular fluorescence complementation (BiFC) assays strongly confirmed the interaction between RSV P2 and OsNPR1 protein *in planta* (Fig. 1b, c). Given the known effects of SA on the function of NPR1, we next investigated the influence of SA on the P2-OsNPR1 interaction. As shown in Fig. 1a, addition of SA did not dramatically enhance the interaction between P2 and OsNPR1. Co-IP experiments were then performed in *NahG* transgenic tobacco plants. The *NahG* gene encodes a salicylate hydroxylase which degrades SA into SAR-inactive catechol and transgenic plants expressing the *NahG* gene do not accumulate SA[39,40]. SA deficiency did not affect the association of P2 and OsNPR1 in *NahG* transgenic plants (Fig. 1d). In addition, BiFC assays also showed that P2 interacts with OsNPR1 protein in *NahG* transgenic plants (Supplementary Fig. 1). These results indicate that P2 specifically interacts with OsNPR1 in a SA-independent manner.

### P2 influences the formation of OsNPR1 oligomers

NPR1 protein is mainly localized in the cytosol as oligomers but when pathogens induce elevated levels of SA, NPR1 turns oligomers into monomers that can enter the nucleus[30,41]. To determine whether P2 protein influences the formation of OsNPR1 oligomers, we next performed BiFC assays using OsNPR1-cYFP/nYFP-OsNPR1 with or without P2-FLAG. As shown in Supplementary Fig. 2a, the recombined YFP signal was strongly observed when OsNPR1-cYFP and nYFP-OsNPR1 were co-expressed in tobacco leaves, but this fluorescence was discernably weaker in the presence of P2-FLAG. Interestingly, we observed that OsNPR1 protein was markedly reduced in *N. benthamiana* leaves co-expressing P2-FLAG (Supplementary Fig. 2b). And we also examined the specificity of OsNPR1 antibody using western blotting assays (Supplementary Fig. 2c, d). To exclude the effect of protein degradation on OsNPR1 oligomers, a 26 S proteasome inhibitor, MG132, was employed in BiFC assays. The results showed that OsNPR1 oligomers were also reduced by P2-FLAG in the presence of MG132 (Supplementary Fig. 2a, b). In addition, we further used an in vitro pull-down assay to confirm the effect of P2 on the formation of oligomers of OsNPR1. The same amounts of GST-OsNPR1 and MBP-HIS-OsNPR1 protein were mixed with different amounts (1x or 10x) of MBP-HIS-P2 or MBP-HIS (control) and subsequently immobilized onto anti-GST beads. Immunoblot analysis showed that the amounts of MBP-OsNPR1 protein bound to GST-OsNPR1 protein decreased with increasing amounts of MBP-P2 (Fig. 1e). Collectively, these results suggest that P2 does indeed interfere with OsNPR1 oligomerization.

### P2 mediates OsNPR1 degradation by the 26 S proteasome pathway

Since levels of OsNPR1 protein were noticeable decreased in the presence of P2-FLAG (Supplementary Fig. 2b), we wondered whether the stability of OsNPR1 was directly affected by P2 *in planta*. We therefore expressed OsNPR1-GFP with or without P2-FLAG in tobacco leaves, and found a noticeably reduced level of OsNPR1 protein in the presence of P2-FLAG (Fig. 2a). The transcript levels of OsNPR1 were not discernably affected by the presence or absence of P2-FLAG (Supplementary Fig. 3a). Further results showed that P2-mediated degradation of OsNPR1-GFP is blocked by the 26 S proteasome inhibitor MG132 (Fig. 2b). Next, we analyzed the OsNPR1 protein levels in rice plants overexpressing OsNPR1 in the presence of P2 protein. *OsNPR1-OX*, a transgenic plant overexpressing *OsNPR1*[42], was crossed with *P2-OX* (a transgenic plant expressing P2 with a FLAG tag)[8], to generate OsNPR1/P2 hybrid plants (named *OsNPR1-OX/P2-OX*). The transcript expression levels of *OsNPR1* in *OsNPR1-OX*, and *OsNPR1-OX/P2-OX* plants were similar and about 100-fold higher than in wildtype TP309 (Supplementary Fig. 3b). The amounts of OsNPR1 protein were evidently less in *OsNPR1-OX/P2-OX* transgenic plants than in *OsNPR1-OX* (Fig. 2c),

indicating that P2 accelerates the degradation of OsNPR1 protein in rice plants. In addition, we also detected the levels of OsNPR1 in *P2-OX* transgenic and WT plants by RT-qPCR and western blotting assays. The results further support the conclusion that P2 promotes the degradation of OsNPR1 protein. (Supplementary Fig. 3c, d). Similarly, in a cell-free degradation system, the degradation rate of OsNPR1 was much higher in the presence of GST-P2 than that in the control GST (Fig. 2d). Thus, we conclude that P2 protein promotes the degradation of OsNPR1 in rice.

To further investigate how P2 influences OsNPR1 degradation, we first analyzed the levels of monomeric and oligomeric OsNPR1 in *OsNPR1-OX/P2-OX* rice plants. OsNPR1, and especially the oligomeric form, was distinctly reduced in the presence of P2 (Fig. 2e). Because P2-mediated degradation of OsNPR1-GFP can be blocked by the 26 S proteasome inhibitor, we next studied whether P2 affects OsNPR1

ubiquitination. Poly-ubiquitination was clearly enhanced in *OsNPR1-OX/P2-OX* transgenic rice compared to *OsNPR1-OX* plants (Fig. 2f), and the poly-ubiquitination of OsNPR1-GFP was markedly enhanced by P2 in *N. benthamiana* leaves (Supplementary Fig. 4a). To investigate the cellular location where viral protein P2 promotes OsNPR1 degradation, we performed western blotting analysis using the nuclear and cytoplasmic fractions of *OsNPR1-OX* and *OsNPR1-OX/P2-OX* samples (Supplementary Fig. 4b). P2 visibly decreased the level of OsNPR1 in the cytoplasm, whereas there was only a small reduction in the nucleus. We further tested the ubiquitination level of OsNPR1 in the different fractions. Poly-ubiquitination of OsNPR1 in the cytoplasm increased visibly with P2 protein, but the poly-ubiquitination in the nucleus remained unchanged (Supplementary Fig. 4c). The results, therefore, suggest that the degradation of OsNPR1 promoted by P2 occurs mainly

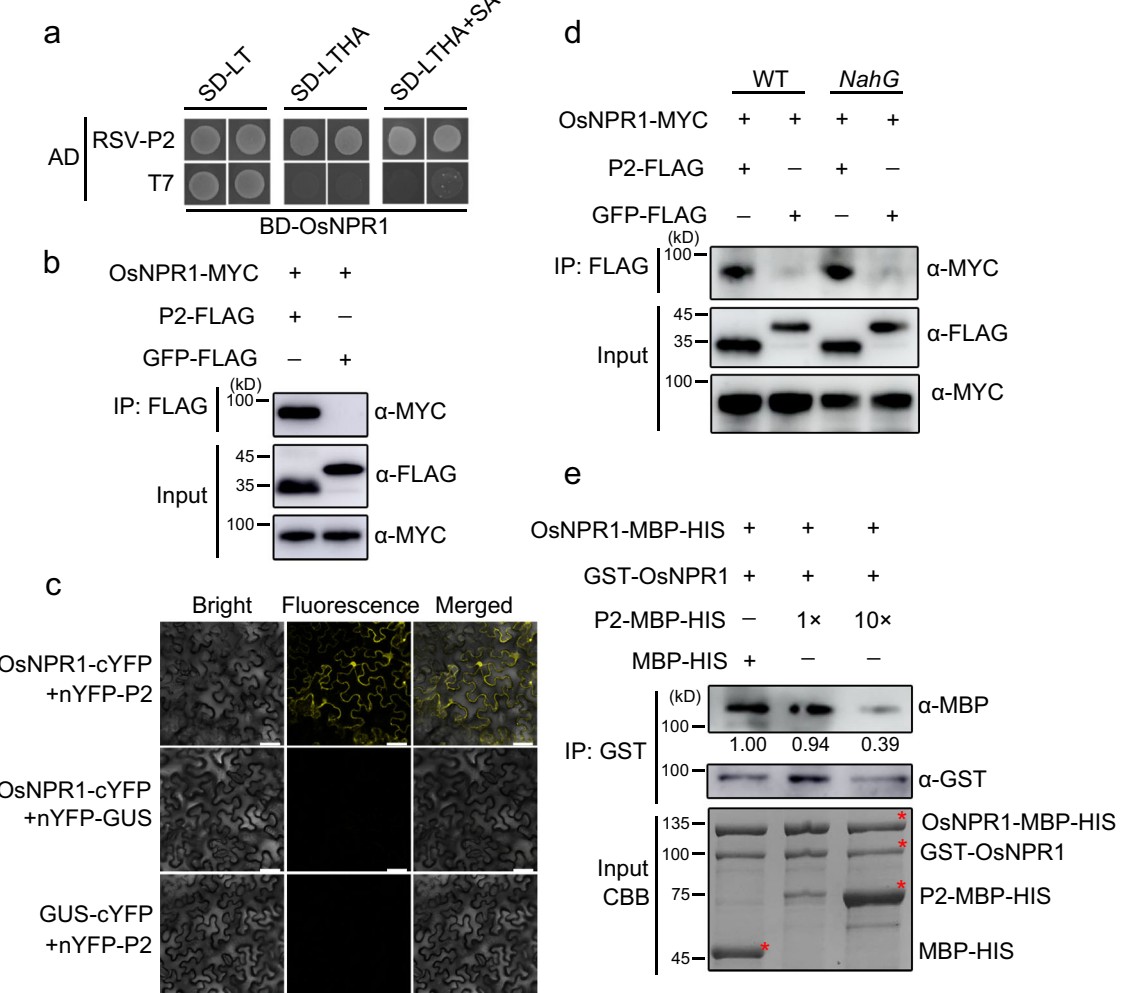

**Fig. 1 | SA-independent interactions between RSV P2 protein and OsNPR1. a** Y2H assay showing the interaction between OsNPR1 and RSV P2 protein in yeast cells. OsNPR1 protein was fused with BD while RSV P2 was fused with AD yeast vectors. The different combinations were transformed into yeast cells and grown on SD-L-T plates at 30 °C for 3 days. Colony growth was scanned after 3 days of incubation in SD-L-T-H-Ade medium with or without SA (0.1 mM). **b, d** Co-immunoprecipitation (Co-IP) assays showing that OsNPR1 interacted with viral protein P2 in *Nicotiana benthamiana* leaves. OsNPR1-MYC and P2-FLAG or GFP-FLAG (negative control) were transiently co-expressed in wildtype (WT) (b) or *NahG* (d) transgenic *N. benthamiana* leaves. Total proteins were extracted, the supernatant precipitated with FLAG beads, and subjected to Co-IP. The immunoprecipitated (IP) and input proteins were analyzed using anti-FLAG and anti-MYC antibodies. **c** BiFC assays confirming the interactions between OsNPR1 and RSV P2 protein in *N. benthamiana*

leaves. OsNPR1-cYFP co-expressed with nYFP-P2 or the negative controls were agro-injected into *N. benthamiana* leaves. The images were captured by confocal microscopy at 48 h post inoculation (hpi). Scale bar = 50 μm. **e** Pull-down assays to assess P2 interference with the OsNPR1 oligomers in vitro. Equal amounts of purified MBP-HIS-OsNPR1 and GST-OsNPR1 were mixed with increasing amounts of MBP-HIS-P2 (1x or 10x) or MBP-His (negative control) in vitro. The interaction between MBP-HIS-OsNPR1 and GST-OsNPR1 was reduced by MBP-HIS-P2. MBP-HIS-OsNPR1 proteins were used to pull down with GST-OsNPR1, and the proteins were analyzed using anti-GST and anti-MBP antibodies. The loading proteins are shown by CBB. Bands shown in figures are indicated by red asterisk. Experiments in (**b**), (**d**) and (**e**) were repeated three times with the similar results. Source data including uncropped scans of gels (**b**), (**d**) and (**e**) are provided in the Source data file.

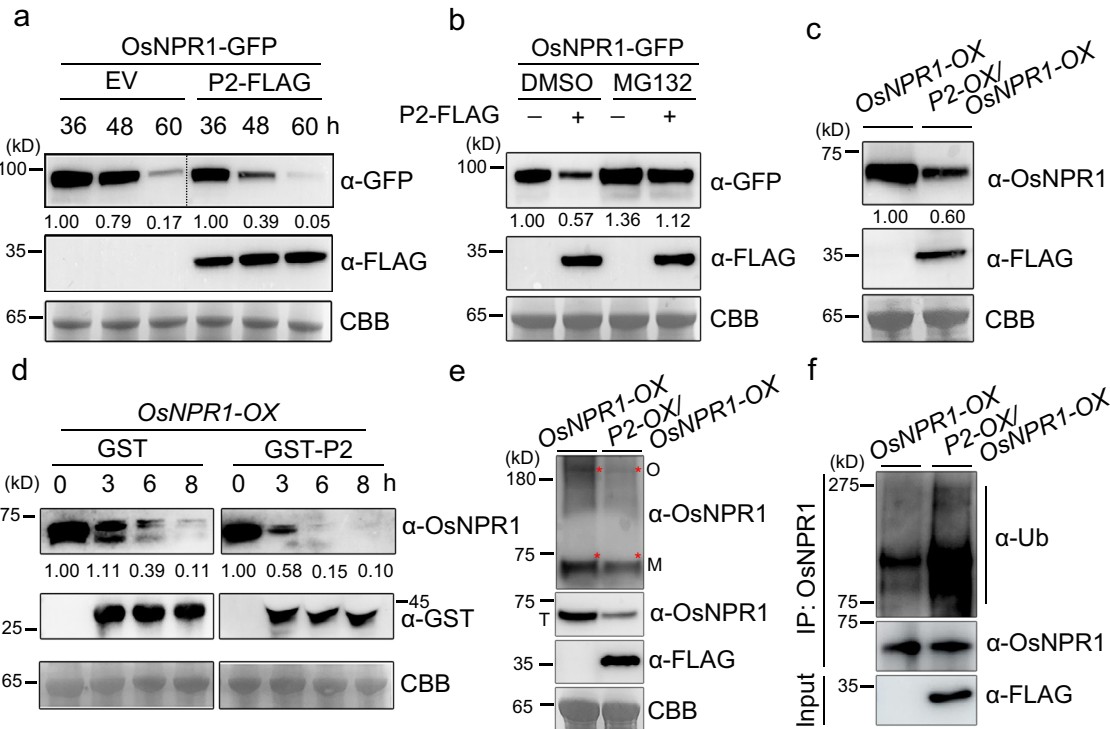

**Fig. 2 | P2 mediates OsNPR1 degradation by the 26 S proteasome pathway. a** P2 triggers NPR1 degradation in *N. benthamiana* leaves. OsNPR1-GFP was co-expressed with or without P2-FLAG in *N. benthamiana* by agroinfiltration and then the extracts were obtained for western blotting at 36, 48 and 60 hpi. CBB was used as a loading control to monitor input protein amounts. **b** P2-triggered OsNPR1 degradation was blocked by proteasome inhibitor MG132. OsNPR1-GFP was co-expressed with or without P2-FLAG in *N. benthamiana* by agroinfiltration and treated with MG132 (100 μM) or DMSO at 24 hpi. The extracts were isolated after treatment with MG132 for western blotting at 24 hpi. CBB was used as a loading control to monitor input protein amounts. **c** Endogenous OsNPR1 protein levels in *OsNPR1-OX* and *OsNPR1-OX/P2-OX* hybrid plants. *P2-OX*, a transgenic plant expressing P2 with a FLAG tag. Samples were taken from 10-day-old seedlings, total proteins were extracted and then immunoblotted by gel blot with anti-OsNPR1 and anti-FLAG antibody. CBB was used as a loading control to monitor input protein amounts. **d** Time course of degradation of OsNPR1 with or without P2 in *OsNPR1-OX* transgenic plants. In vitro degradation assay, the plant crude extracts from *OsNPR1-OX* seedlings were incubated with 50 μg purified GST or GST-P2 purified proteins at 37 °C, and samples

were collected at the indicated times for western blot using anti-OsNPR1 and anti-GST. CBB was used as a loading control to monitor input protein amounts. **e** P2 reduces the accumulation of oligomeric and monomeric OsNPR1. The nonreduced protein extracts of *OsNPR1-OX/P2-OX* and *OsNPR1-OX* rice leaves were obtained for western blotting at 48 hpi. *P2-OX*, a transgenic plant expressing P2 with a FLAG tag. The proteins were separated by non-reducing and reducing SDS-PAGE gels by immunoblot using anti-OsNPR1 and anti-FLAG antibodies. O: OsNPR1 oligomeric; M: OsNPR1 monomeric; Bands shown in figures are indicated by red asterisk. T: total proteins. CBB was used as a loading control to monitor input protein amounts. **f** The protein extracts of *OsNPR1-OX/P2-OX* and *OsNPR1-OX* rice leaves were extracted with IP lysis buffer containing 100 μM MG132 and 10 mM DTT and precipitated with Protein A/G OsNPR1 antibody beads. *P2-OX*, a transgenic plant expressing P2 with a FLAG tag. Similar amounts of OsNPR1 precipitated by the antibody beads were used for analysis by immunoblotting using anti-Ubiquitin (Ub), anti-OsNPR1 and anti-FLAG antibodies. Experiments in (**a**)–(**f**) were repeated three times with the similar results. Source data including uncropped scans of gels (**a**–**f**) are provided in the Source data file.

in the cytoplasm. Collectively, these results showed that P2 promoted the degradation of OsNPR1 via the host 26 S proteasome *in planta*.

### P2 promotes the association between OsCUL3a and OsNPR1

Since P2 interacts with OsNPR1 in a SA-independent manner, we wondered whether SA influences OsNPR1 destabilization mediated by P2. We, therefore, expressed OsNPR1-GFP with or without P2-FLAG in WT and *NahG* tobacco leaves. As in WT plants, the levels of OsNPR1 protein in *NahG* plants were evidently reduced in the presence of P2-FLAG (Fig. 3a), indicating that the destabilization of OsNPR1 by P2 was independent of SA. Previous studies have shown that adapters Cullin3 (CUL3) can interact with NPR family members for ubiquitination and degradation[35,43]. In rice, OsCUL3a was shown to interact with RING-BOX1 (OsRBX1a and OsRBX1b) to form a CUL-Ring-like E3 ubiquitin ligase complex, which accelerated the degradation of OsNPR1 through the 26 S proteasome[44]. Since P2 promotes the degradation of OsNPR1 by the 26 S proteasome pathway, we speculated that P2 might affect the association of OsNPR1 and OsCUL3a. We, therefore, tested the interaction between OsNPR1 and OsCUL3a in *N. benthamiana* leaves followed by Co-IP. The results showed that

the OsNPR1-OsCUL3a interaction was very weak in the absence of SA, but much stronger in the presence of SA (Fig. 3b). Consistent with previous results that SA promoted the association of NPR1 with CUL3a in *Arabidopsis*[43], our results confirmed that SA promotes the OsNPR1-OsCUL3a interaction. OsNPR1 protein contains an N-terminal BTB/POZ domain, a middle ankyrin repeat (ANK) domain and a C-terminal domain[29,33]. We next constructed different OsNPR1 truncations based on its conserved domains to study by Co-IP assays which regions were responsible for theOsNPR1-OsCUL3a interaction. In contrast to the full-length protein, deleting the C-terminal domain (CTD) of OsNPR1 (ΔCTD) strikingly enhanced the association with OsCUL3a even in the absence of SA (Fig. 3c). The results also show that the BTB domain of OsNPR1 is required for its interaction with OsCUL3a (Fig. 3c). Similarly, a LCI assay showed that the BTB domain was associated with OsCUL3a (Supplementary Fig. 5c, e, f). In addition, we found that the BTB domain directly interacted with the CTD domain by Co-IP and LCI assays (Fig. 3d and Supplementary Fig. 5d). A protein competition Co-IP assay was used to show that the CTD domain outcompeted OsCUL3a for interaction with the BTB domain (Fig. 3e). These results indicate that the CTD domain blocks the

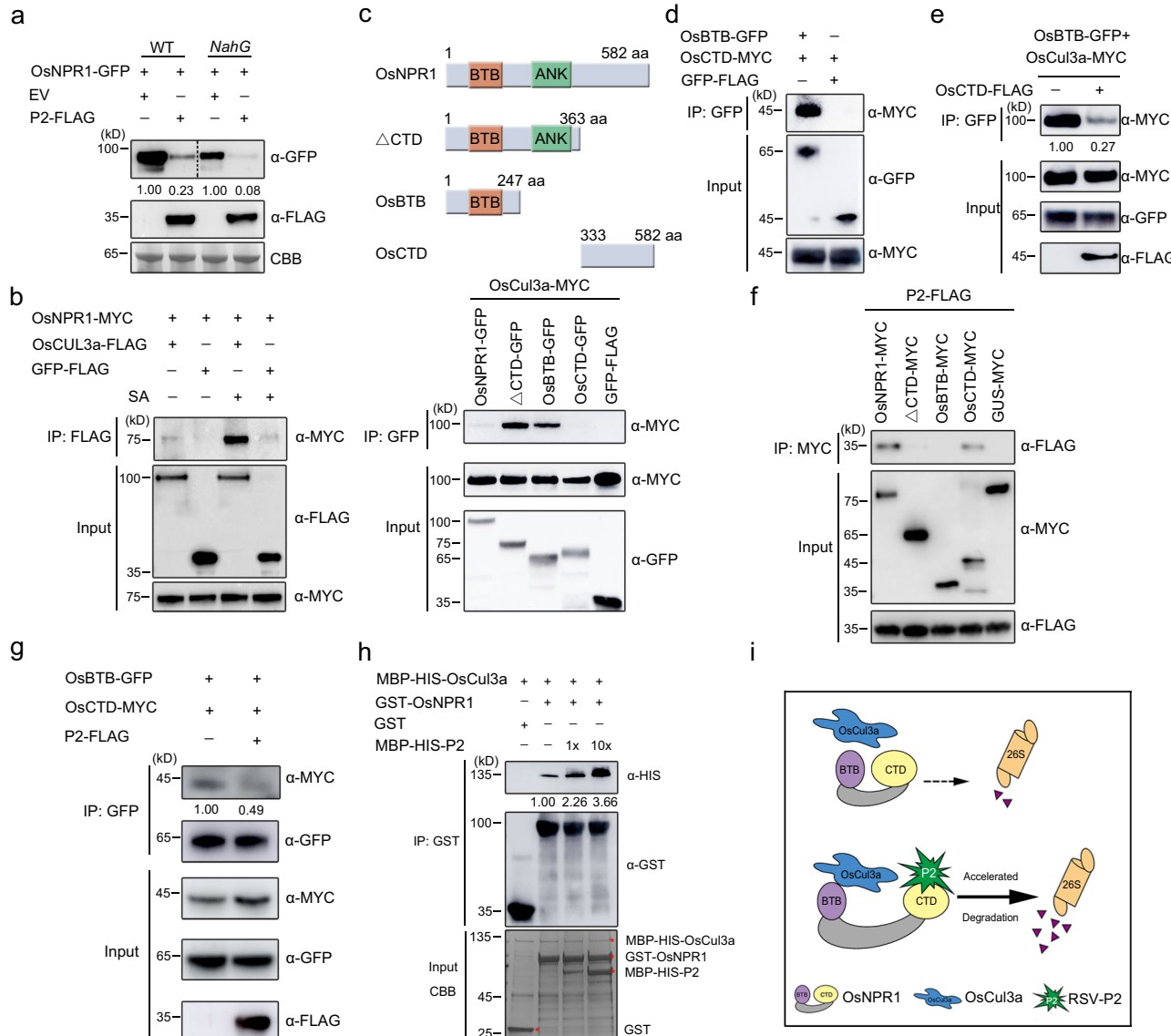

**Fig. 3 | P2 promotes the association between OsCUL3a and OsNPR1. a** P2-triggered OsNPR1 degradation was independent of SA. OsNPR1-GFP and P2-FLAG or EV (empty vector) were transiently co-expressed in WT or *NahG* transgenic *N. benthamiana* leaves. The extracts were isolated for western blotting using anti-GFP and anti-FLAG antibodies at 48 hpi. CBB was used as a loading control to monitor input protein amounts. **b** Co-IP analyses of the interactions between OsNPR1 and OsCUL3 in *N. benthamiana* leaves treated with water or 500 μM SA for 5 h. OsNPR1-MYC and OsCUL3a-FLAG or GFP-FLAG (negative control) were transiently co-expressed in tobacco leaves. Total proteins were extracted, the supernatant precipitated with FLAG beads and the immunoprecipitated (IP) and input proteins were then analyzed using anti-FLAG and anti-MYC antibodies. **c** Schematic diagrams of OsNPR1 and its deletion mutations used to test interactions with the OsCUL3a protein (upper). Co-IP analyses showing the interaction of OsCUL3 with OsNPR1 variants in *N. benthamiana*. OsCUL3a-MYC and OsNPR1-GFP deletion mutations or GFP-FLAG (negative control) were transiently co-expressed in tobacco leaves (lower). Total proteins were extracted and the supernatant precipitated with GFP beads. **d** Co-IP analyses of the interactions between BTB and CTD domain of OsNPR1 in *N. benthamiana*. OsCTD-MYC and OsBTB-GFP or GFP-FLAG (negative control) were transiently co-expressed in tobacco leaves. Total proteins were

extracted and the supernatant precipitated with GFP beads. **e** The CTD domain of OsNPR1 outcompeted OsCUL3a for interaction with the BTB domain *in planta*. OsBTB-GFP and OsCUL3a-MYC were transiently infiltrated with/without OsCTD-FLAG. **f** Co-IP assays showing the association of P2 with OsNPR1 deletion mutants in *N. benthamiana*. OsNPR1-MYC, OsBTB-MYC, OsCTD-MYC, P2-FLAG and GFP-FLAG (negative control) were transiently co-expressed in tobacco leaves. Total proteins were extracted and the supernatant precipitated with GFP beads. **g** Co-IP assays showing that P2 reduced the interaction between OsBTB and OsCTD *in planta*. OsBTB-GFP and OsCTD-MYC were transiently infiltrated with/without P2-FLAG. **h** Pull-down assays to test P2 interference with the interaction between OsNPR1 and OsCUL3a in vitro. Equal amounts of purified MBP-HIS-OsCUL3a and GST-OsNPR1 were mixed with increasing amounts of MBP-HIS-P2 (1x or 10x) in vitro. The interaction between OsNPR1 and OsCUL3a was reduced by MBP-HIS-P2. The loading of MBP-HIS-OsCUL3A, GST-OsNPR1, MBP-HIS-P2 and GST were shown by CBB. **i** Schematic diagram illustrating how P2 promotes the association between OsCUL3a and OsNPR1 to influence the degradation of OsNPR1. Experiments in **a**–**h** were repeated three times with the similar results. Source data including uncropped scans of gels (**a**–**h**) are provided in the Source data file.

association of OsNPR1 with OsCUL3a by directly competing with OsCUL3a for binding to the BTB domain.

　　We then used the truncated mutants of OsNPR1 to demonstrate that P2 strongly interacted with the CTD domains in Co-IP and Y2H

assays (Fig. 3f and Supplementary Fig. 5a, b). Because P2 specifically associates with the CTD domain of OsNPR1, it seemed likely that P2 would inhibit the ability of CTD to outcompete OsCUL3a for interaction with the BTB domain and therefore accelerate the association of

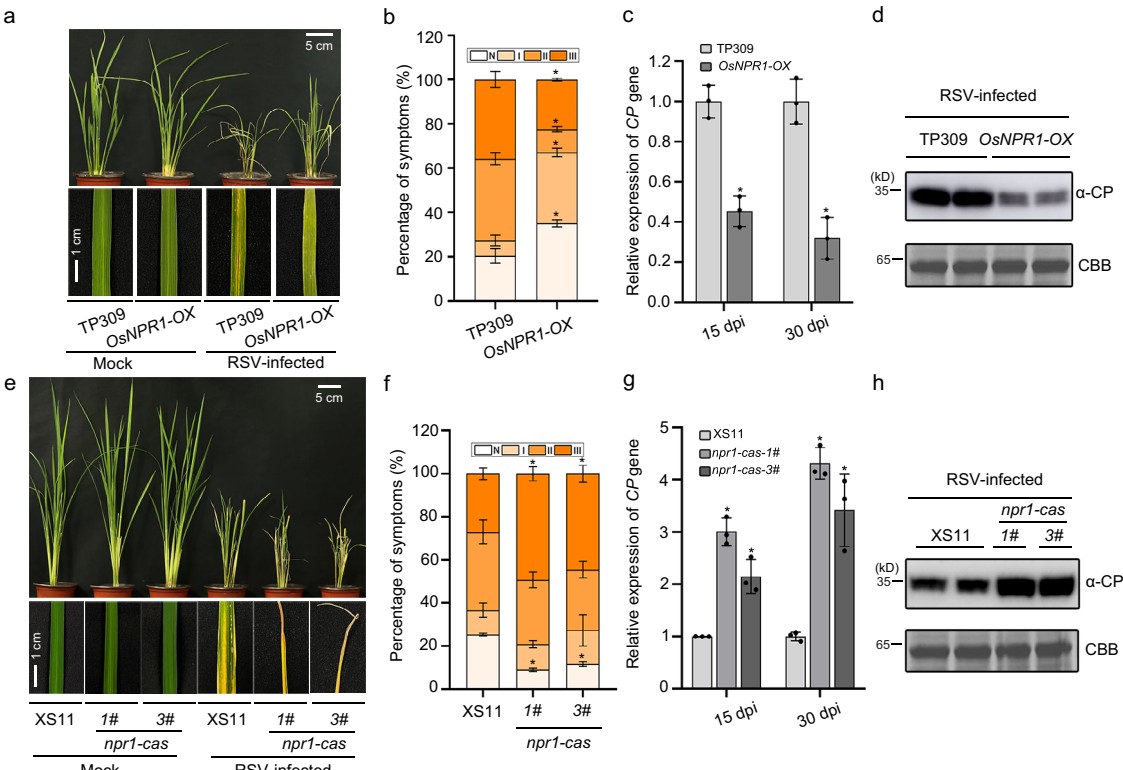

**Fig. 4 | OsNPR1 confers resistance to RSV infection in rice. a** Viral symptoms on TP309 and *OsNPR1-OX* in response to RSV infection. The phenotypes were observed and photos taken at 30 dpi. Scale bars = 5 cm or 1 cm. **b** The disease incidence in TP309 (*n* = 23) and *OsNPR1-OX* (*n* = 35) and the different grades of disease symptoms. The percentage of plants infected by RSV was determined by RT-PCR at 30 dpi. Error bars represent SD, values are means ± SD (*n* = 3 biologically independent replicates per genotype). Significant differences were analyzed using one-way ANOVA followed by Tukey's multiple comparisons test. * at the columns indicate significant differences (*p* ≤ 0.05). **c** The relative mRNA levels of RSV *CP* in RSV-infected TP309 and *OsNPR1-OX* rice plants as detected by RT-qPCR at 15 or 30 dpi. Error bars represent SD, values are means ± SD (*n* = 3 biologically independent replicates per genotype). Significant differences were analyzed using one-way ANOVA followed by Tukey's multiple comparisons test. * at the columns indicate significant differences (*p* ≤ 0.05). **d** The accumulation of RSV CP protein in RSV-infected TP309 and *OsNPR1-OX* rice plants determined by western blotting. CBB serves as the loading control to monitor input protein amounts. **e** Typical symptoms on XS11 (control), *npr1-cas-1#* and *npr1-cas-3#* 30 d after RSV

inoculation. Scale bars = 5 cm or 1 cm. **f** The disease incidence in XS11 (*n* = 20), *npr1-cas-1#* (*n* = 20) and *npr1-cas-3#* (*n* = 19) with different grades of disease symptoms. The percentage of plants infected by RSV was determined by RT-PCR at 30 dpi. Error bars represent SD, values are means ± SD (*n* = 3 biologically independent replicates per genotype). Significant differences were analyzed using one-way ANOVA followed by Tukey's multiple comparisons test. * at the columns indicate significant differences (*p* ≤ 0.05). **g** Relative mRNA levels of RSV *CP* gene detected by RT-qPCR in RSV-infected XS11, *npr1-cas-1#* and *npr1-cas-3#* rice plants at 15 or 30 dpi. Error bars represent SD, values are means ± SD (*n* = 3 biologically independent replicates per genotype). Significant differences were analyzed using one-way ANOVA followed by Tukey's multiple comparisons test. * at the columns indicate significant differences (*p* ≤ 0.05). **h** The accumulation of RSV CP protein in RSV-infected XS11, *npr1-cas-1#* and *npr1-cas-3#* rice plants determined by western blotting. CBB serves as the loading control to monitor input protein amounts. Experiments in (**d**) and (**h**) were repeated three times with the similar results. Source data including disease incidence (**b** and **f**) uncropped scans of gels (**d** and **h**) and *p* values of statistic tests (**b**, **c**, **f**, and **g**) are provided in the Source data file.

OsNPR1 with OsCUL3a. To test our hypothesis, we performed a protein competition Co-IP assay in tobacco and found that P2 indeed impaired the interaction between the BTB and CTD domains (Fig. 3g). To confirm that P2 promotes the interaction between OsNPR1 and OsCUL3a, we performed a pull-down assay in vitro and observed an increased interaction between OsNPR1 and OsCUL3a with increasing amounts of MBP-P2 (Fig. 3h). Together, these results support our hypothesis that P2 interferes with the interaction between the CTD and BTB domains and promotes the association of OsCUL3a and OsNPR1, resulting in the acceleration of OsNPR1 degradation (Fig. 3i).

### OsNPR1 positively modulates rice antiviral defense

The ability of viral protein P2 to accelerate the degradation of OsNPR1 prompted us to explore the effect of OsNPR1 on viral infection. Rice plants overexpressing OsNPR1 (line *OsNPR1-OX*) and its wildtype TP309 were inoculated with RSV. RSV infection caused the typical leaf yellow stripes symptoms in wildtype plants, but symptoms were milder with discontinuous yellow stripes in the *OsNPR1-OX* plants (Fig. 4a). *OsNPR1-OX* plants had distinctly less severe stunting about

30 dpi after RSV inoculation, and lower percentages of typical disease symptoms (grade II and grade III) compared with TP309 plants (Fig. 4b, Supplementary Fig. 6 and Supplementary Table 2). RT-qPCR and western blotting analysis also showed that the transcription and protein levels of RSV coat protein (CP) were reduced in the line overexpressing *OsNPR1* than in the control plants (Fig. 4c, d). To confirm these results, we further constructed lines overexpressing OsNPR1 in a NIP background, and selected two homozygous OsNPR1-overexpressing lines (named *OsNPR1-2#* and *OsNPR1-7#*) for viral inoculation (Supplementary Fig. 7a). The results showed that *OsNPR1* plants were more resistant to RSV infection than NIP plants (Supplementary Fig. 7b–d). These results indicated that overexpression of OsNPR1 contributes to rice resistance to RSV infection.

We next used two types of CRISPR/Cas9 mutant lines *npr1-cas-1#* and *npr1-cas-3#* to study the role of OsNPR1 in rice antiviral defense. RSV-infected *npr1-cas* mutants had more severe curling or death of the young leaves and had higher percentages of plants with severe disease symptoms (grade III) compared with control plants (Fig. 4e, f and Supplementary Table 2). The RNA and protein of RSV CP also

accumulated to noticeably higher levels in *npr1-cas-1#* and *npr1-cas-3#* lines than in WT plants (Fig. 4g, h). Together, these data indicate that OsNPR1 plays a positive role in antiviral immunity against RSV.

## OsNPR1 triggered JA signaling by disturbing the OsJAZ-OsMYC2 complex

The SA and JA signaling pathways are two essential defense hormones, and their cross talk determines the outcome of plant immunity in response to different plant pathogens[45,46]. Our recent work showed that RSV P2 protein negatively modulated JA signaling by cooperating with OsJAZ repressors to repress the transcriptional activation of OsMYC2/3 transcription factors[8]. Because we have now shown that P2 also directly interacts with OsNPR1, we hypothesized that OsNPR1 might be directly associated with the JA signaling key components (OsJAZ proteins or OsMYC2/3 factors). To test this hypothesis, we initially used OsNPR1 as bait to screen the interaction with these proteins by Y2H assay. The 15 OsJAZ proteins except OsJAZ2 (OsJAZ1, OsJAZ3-15) were cloned and tested for any interaction with OsNPR1. OsJAZ5, OsJAZ9, and OsJAZ11 all interacted with OsNPR1 (Supplementary Fig. 8a). To define the interaction between OsNPR1 and OsJAZs, the OsNPR1-MYC was expressed together with OsJAZ9-FLAG or GFP-FLAG (negative control) in tobacco leaves, and the Co-IP experiment was conducted in vivo. OsNPR1-MYC was coimmunoprecipitated by OsJAZ9-FLAG, but not by the negative GFP-flag (Fig. 5a). Similar results from BiFC assays showed that OsNPR1 interacts with OsJAZ9 *in planta* (Supplementary Fig. 8b). Together, these results suggested that OsNPR1 indeed interacts with OsJAZ proteins.

Secondly, we detected whether OsNPR1 directly interacts with JA signaling key transcription factors OsMYC2/3. A Co-IP assay showed that OsNPR1 specifically interacted with both OsMYC2 and OsMYC3 (Fig. 5b) and this was further confirmed by BiFC (Supplementary Fig. 8c). Given that OsNPR1 interacted with both OsJAZ and OsMYC2/3, we then tested whether OsNPR1 directly influences the association of OsJAZ9 with OsMYC2/3. We co-expressed OsMYC2-cYFP/OsMYC3-cYFP and nYFP-OsJAZ9 in tobacco leaves with or without OsNPR1-MYC. The results showed that the fluorescence formed by OsMYC2/3-cYFP and nYFP-OsJAZ9 was evidently reduced in the presence of OsNPR1-MYC (Supplementary Fig. 9a, b). In a protein competition Co-IP assay, the association between OsMYC2/3 and OsJAZ9 was markedly decreased in the presence of OsNPR1 protein in *N. benthamiana* (Fig. 5c and Supplementary Fig. 9c). In further support of our results, we performed competitive Co-IP assays using *npr1-cas* mutant rice plants. The results showed that OsJAZ9-OsMYC2 interaction was enhanced in the absence of OsNPR1 protein in rice (Supplementary Fig. 9d–f). Together, these results suggested that OsNPR1 interrupted the OsJAZ9-OsMYC2/3 interaction.

Because OsNPR1 is a transcriptional activator[34,47,48], we wondered whether OsNPR1 affected the transcriptional activity of OsMYC2/3. Recent reports indicated that OsMYC2 could bind to the promoters of the *OsMADS1* and *OsNOMT* genes[49,50]. Therefore, we fused the promoters of *OsMADS1* and *OsNOMT* with a firefly luciferase (LUC) to construct the *pOsMADS1::LUC* and *pOsNOMT::LUC* vectors for use in a dual-luciferase transcriptional activity assay (Fig. 5d, upper). The transcriptional activity of OsMYC2 was markedly increased in the presence of OsNPR1 (Fig. 5d, lower). To investigate the activation of JA-regulated gene expression by OsNPR1 in vivo, we conducted chromatin immunoprecipitation (ChIP) qPCR (ChIP-qPCR) on OsNPR1-overexpressing (*OsNPR1-7#*) plants. Firstly, we performed ChIP-qPCR using OsMYC2-specific polyclonal antibodies in NIP or *Ri-m2m3* mutant plants, in which the expression of *OsMYC2* and *OsMYC3* were significantly decreased (Supplementary Fig. 10a), to confirm that OsMYC2 specifically binds to the promoters of the *OsMADS1* and *OsNOMT* genes in vivo (Supplementary Fig. 11a, b). We then performed ChIP-qPCR in NIP and *Ri-m2m3* mutant plants using OsNPR1-specific polyclonal antibodies. The results showed that OsNPR1 specifically

bound to the G-box motif in the promoters of *OsMADS1* and *OsNOMT* in wildtype NIP. However, the ability of OsNPR1 to bind to these promoters of *OsMADS1* and *OsNOMT* was significantly decreased in *Ri-m2m3* mutant plants (Fig. 5e, f). These results indicated that OsNPR1 is recruited to the promoter region of JA-regulated genes by associating with OsMYC2. Thirdly, we performed ChIP-qPCR assays on OsNPR1-overexpressing (*OsNPR1-7#*) plants. The results showed that the promoters of JA-responsive genes enriched by OsNPR1 were increased in OsNPR1-overexpressing plants than NIP plants (Fig. 5g). In addition, the expression levels of *OsMADS1* and *OsNOMT* genes were significantly enhanced in OsNPR1-overexpressing plants compared to NIP plants (Supplementary Fig. 11c). These results further indicate that OsNPR1 activates JA-responsive genes by forming a complex with OsMYC2.

To further investigate the biological significance of OsNPR1 in JA signaling, we used the OsNPR1 transgenic rice plants to analyze JA sensitivity. JA treatment usually inhibits root growth in plants, and this inhibitory effect is enhanced in plants where JA signaling is activated[8,49]. We treated the seedling roots of *OsNPR1-OX* and its wildtype with different concentrations of MeJA for 5 days in the dark. As expected, the root growth of WT plants was discernably inhibited by 0.5 or 1 μM MeJA treatment (Supplementary Fig. 12a) and this inhibitory effect was notably enhanced in transgenic plants overexpressing *OsNPR1* (Supplementary Fig. 12a, b), indicating that overexpression of *OsNPR1* in rice plants activated JA signaling. Together, these observations suggested that OsNPR1 activated JA signaling by directly promoting the DNA binding activity of OsMYC2.

In contrast to the well-known SA-JA antagonism in *Arabidopsis*[45,46], our results here show that OsNPR1 activated JA signaling in rice. To further investigate the relationship between SA and JA, we treated rice seedling roots with SA and/or JA. There was a slight reduction in the root length of WT plants following 1 μM SA treatment (Supplementary Fig. 12c–f), but when a mixture of 0.5 μM MeJA and 1 μM SA was applied, the root length was severely reduced compared to MeJA treatment alone (Supplementary Fig. 12c–f). In contrast to the WT plants, exogenous application of SA had no significant difference in *npr1-cas* mutant lines. In addition, the JA-SA synergistic inhibition of root length in *npr1-cas-1#* and *npr1-cas-3#* was markedly less than in WT plants (Supplementary Fig. 12c, d). These results indicated that the synergistic effect of SA on activating JA signaling was dependent on OsNPR1.

We next investigated the contribution of JA signaling key transcription factors OsMYC2/3 to the SA-JA interaction. Two homozygous OsMYC2/OsMYC3 mutants, *Ri-m2m3-4#* and *Ri-m2m3-6#*, were used. When the *Ri-m2m3* mutants were treated with a mixture of 0.5 μM MeJA and 1 μM SA, there was some synergistic inhibition of root length but decidedly less than in WT plants (Supplementary Fig. 12e, f). These findings suggested that the SA-mediated activation of JA signaling and the synergistic effect of SA-JA on JA signaling were dependent on OsMYC2/3. Together, OsNPR1 is indispensable for the SA-mediated activation of JA signaling, which is also dependent on JA signaling key transcription factors OsMYC2/3.

To further clarify the role of OsNPR1 in JA signaling, we crossed plants overexpressing *OsNPR1* with *OsMYC2/3* RNAi mutants, and obtained homozygous *OsNPR1-7#/Ri-m2m3-6#* hybrid plants. We then assessed the sensitivity of the hybrid plants to RSV infection. Like *Ri-m2m3* plants, *OsNPR1-7#/Ri-m2m3-6#* hybrid plants were hypersensitive to RSV infection compared with wildtype and plants overexpressing *OsNPR1* (Fig. 5h–j and Supplementary Fig. 10b, c). Together, these results suggest that OsNPR1-mediated antiviral defense largely depends on JA signaling key transcription factors OsMYC2/3.

## P2 inhibits the interaction between OsNPR1 and OsMYC2/3

Since both OsNPR1 and OsMYC2/3 were directly targeted by viral protein P2, we then tested whether P2 affected the association between OsNPR1 and OsMYC2/3 using a protein competition Co-IP

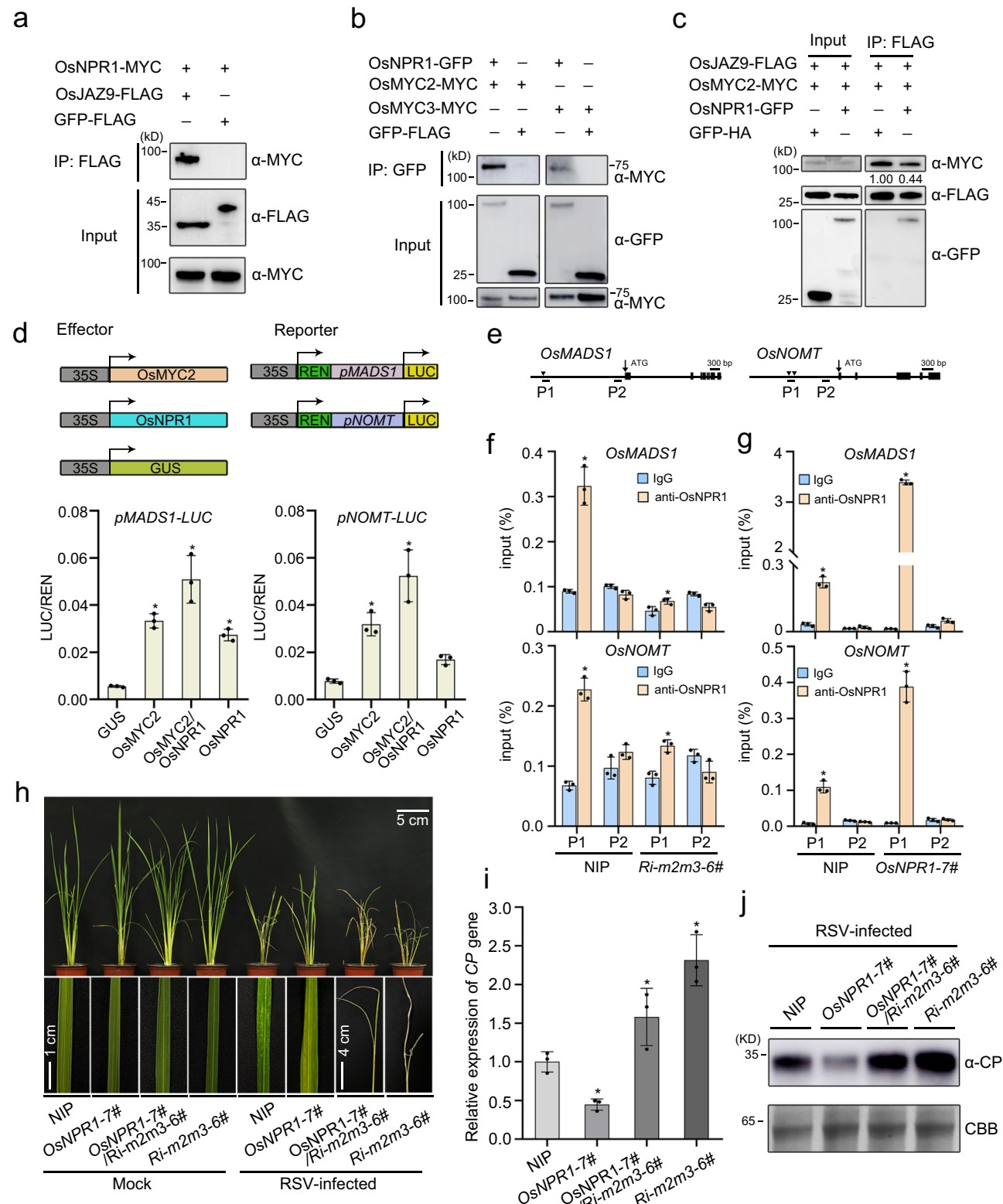

assay in *N. benthamiana* leaves. Because P2 promotes OsNPR1 degradation, MG132 was added in these assays. P2 clearly interfered with the association between OsNPR1 and OsMYC2/3 (Fig. 6a) and this was confirmed in a BiFC assay where the YFP signals formed by nYFP-OsNPR1 and cYFP-OsMYC2/3 was clearly reduced in the presence of P2 (Supplementary Fig. 13).

To explore the biological significance of this P2-mediated disassociation of OsNPR1 and OsMYC2/3, we performed a JA sensitivity

assay using *OsNPR1-OX, OsNPR1-OX/P2-OX* and *P2-OX* transgenic plants with WT controls. The inhibitory effect of MeJA on root length was dramatically enhanced in *OsNPR1-OX* plants but explicitly alleviated in *OsNPR1-OX/P2-OX* plants compared with the WT plants (Fig. 6b–d). Thus, JA-mediated inhibition of root length enhanced by over-expression of OsNPR1 was subverted in the presence of P2 protein. Next, the sensitivity of *OsNPR1/P2* hybrid plants to RSV infection was evaluated. Expression of RSV P2 protein resulted in more severe

**Fig. 5 | OsNPR1 plays a positive role in activating JA signaling. a, b** Co-IP assays showing that OsNPR1 interacts with OsJAZ9, OsMYC2 or OsMYC3. OsNPR1 and OsJAZ9, OsMYC2 and OsMYC3 were transiently co-expressed in tobacco leaves. **c** Protein competition analyzed by Co-IP assays *in planta*. OsJAZ9-FLAG and OsMYC2-MYC or OsMYC3-MYC were infiltrated with or without OsNPR1-GFP in leaves of *N. benthamiana*, HA-GFP serves as negative control. The samples were harvested at 48 hpi for coimmunoprecipitation with FLAG beads. **d** OsNPR1 elevated the transcriptional activation activity of OsMYC2. (Upper) Schematic diagram of the dual-LUC assays. The promoters of *OsMADS1* and *OsNOMT* with a firefly luciferase (LUC) were used to construct the *pOsMADS1::LUC* and *pOsNOMT::LUC* vectors as the reporters. Renilla luciferase (REN) was the internal control. OsNPR1 and OsMYC2 were the effectors. (Lower) The *OsMADS1* and *OsNOMT* promoters were activated by OsMYC2 protein, and this activation were highly enhanced by co-expression with OsNPR1 in *N. benthamiana*. The LUC/REN ratio represents the relative LUC activity. Error bars represent SD, values are means ± SD ($n = 3$ biologically independent replicates per genotype). Significant differences were analyzed using one-way ANOVA followed by Tukey's multiple comparisons test. * at the columns indicate significant differences ($p \le 0.05$). **e** Schematic diagram of *OsMADS1* and *OsNOMT* genes with exons indicated as black boxes for ChIP-qPCR analyses. Black triangles denote the CACGTG (G-box) motifs. Arrowheads denote the transcription start sites. P1 and P2 denote the corresponding amplicons for qPCR. **f** ChIP-qPCR analyses of OsNPR1 binding to the G-box from *OsMADS1* and *OsNOMT* promoters in NIP and *OsNPR1-7#* plants using OsNPR1-specific polyclonal

antibodies. Error bars represent SD, values are means ± SD ($n = 3$ biologically independent replicates per genotype). Significant differences were analyzed using one-way ANOVA followed by Tukey's multiple comparisons test. * at the columns indicate significant differences ($p \le 0.05$). **g** ChIP-qPCR analyses of OsNPR1 binding to the G-box from *OsMADS1* and *OsNOMT* promoters in NIP and *Ri-m2m3-6#* plants using OsNPR1-specific polyclonal antibodies. Error bars represent SD, values are means ± SD ($n = 3$ biologically independent replicates per genotype). Significant differences were analyzed using one-way ANOVA followed by Tukey's multiple comparisons test. * at the columns indicate significant differences ($p \le 0.05$). **h** Viral symptoms in *OsNPR1-7#* (n = 25), *OsNPR1-7#/Ri-m2m3* (n = 22), *Ri-m2m3* (n = 25) transgenic plants and NIP ($n = 21$) in response to RSV infection. The phenotypes were observed and photos taken at 30 dpi. Scale bars = 5 cm, 4 cm or 1 cm. **i** The relative mRNA levels of RSV *CP* in RSV-infected *OsNPR1-7#, OsNPR1-7#/Ri-m2m3-6#, Ri-m2m3-6#* transgenic and NIP rice plants as detected by RT-qPCR at 30 dpi. Error bars represent SD, values are means ± SD ($n = 3$ biologically independent replicates per genotype). Significant differences were analyzed using one-way ANOVA followed by Tukey's multiple comparisons test. * at the columns indicate significant differences ($p \le 0.05$). **j** The accumulation of RSV CP protein in RSV-infected *OsNPR1-7#, OsNPR1-7#/Ri-m2m3-6#, Ri-m2m3-6#* transgenic and NIP rice plants. CBB serves as the loading control to monitor input protein amounts. Experiments in (**a**)−(**c**) and (**j**) were repeated three times with the similar results. Source data including uncropped scans of gels (**a**−**c** and **j**) and *p* values of statistic tests (**d**, **f**, **g** and **i**) are provided in the Source data file.

symptoms than in *OsNPR1-OX* transgenic plants (Fig. 6e). In addition, the RNA and protein levels of RSV CP were greater in *OsNPR1-OX/P2-OX* than in *OsNPR1-OX* transgenic plants (Fig. 6f, g).

To further investigate the relationship between SA- and JA-induced resistance to RSV, seedlings were inoculated with RSV and then sprayed with SA and/or JA and 1% Triton X-100 as control. RT-qPCR experiments showed that the RNA level of RSV CP was slightly reduced in NIP plants following SA treatment, but when a mixture of JA and SA was applied, the accumulation levels of CP were distinctly reduced compared to either SA or JA alone (Fig. 6h). When seedlings of *P2-OX* transgenic plants were treated with SA and/or JA after RSV infection, the JA-SA synergistic antiviral effect was significantly less than in NIP plants (Fig. 6h). Together, these findings suggest that P2 protein directly interferes with OsNPR1-mediated activation of JA signaling and inhibits the SA-JA synergistic triggered antiviral defense.

### OsNPR1-mediated activation of JA signaling is disrupted by other viral effectors

Our previous results showed that RSV P2, SRBSDV SP8, and RSMV M all directly interacted with OsMYC2/3 to impair JA-mediated antiviral immunity[8]. Although SRBSDV SP8 and RSMV M protein did not interact directly with OsNPR1 (Supplementary Fig. 14), we wondered if they could act like RSV P2 and affect the association between OsNPR1 and OsMYC2/3. We therefore conducted protein competition Co-IP assays between OsNPR1 and OsMYC3 in the presence or absence of SP8 and M proteins in *N. benthamiana*. Intriguingly, we found that while OsMYC3-MYC was precipitated by OsNPR1-GFP, this interaction appeared to be reduced in the presence of SP8 or M protein (Fig. 7a, e). We also found that the synergistic effect of JA-SA on root length was obviously alleviated in SP8-*OX* and *M-OX* lines (transgenic rice plants expressing the SP8 or M protein) compared to control NIP plants (Fig. 7b, c, f, g). These results show that SP8 and M protein can also inhibit the synergistic effect of JA and SA signaling.

We further tested the function of OsNPR1 in response to infection by different viruses. When lines overexpressing OsNPR1 (*OsNPR1-2#* and *OsNPR1-7#*) were challenged with SRBSDV, RT-qPCR analysis showed that the levels of SRBSDV RNAs (*S2, S4* and *S6*) were much less than in the control NIP plants (Fig. 7d). Similarly, after inoculation with RSMV, the transgenic plants had significantly lower levels of RSMV *N* gene than the controls (Fig. 7h). These results suggest that OsNPR1 provides broad-spectrum antiviral immunity, not only against the

*tenuivirus* RSV but also against the *fijivirus* SRBSDV and the *cytorhabdovirus* RSMV.

In summary, we show that OsNPR1, a master regulator of SA signaling, displays broad-spectrum antiviral defense against very different rice viruses, including those with dsRNA and ssRNA genomes. OsNPR1 interacts with OsJAZ proteins and OsMYC2/3 transcription factors and then inhibits the association between OsMYC2/3 and OsJAZ to provoke JA signaling. The different types of rice virus encode unrelated viral proteins that disturb the OsNPR1-mediated activation of JA signaling to facilitate viral infection (Fig. 8). Together, our findings reveal a detailed mechanism about the interplay of SA-JA and shed light on a novel strategy in which distinct viral proteins generally repress the interlinking of the JA-SA pathway to subvert rice antiviral immune responses.

## Discussion

The crosstalk between SA and JA, and especially their antagonistic interaction, has been widely reported in studies of plant-pathogen interactions[51-55]. Despite this clear antagonistic action, JA-SA relationships can differ depending on the relative concentration of the two hormones[45] and factors such as developmental stage, environmental stress and pathogen attack can determine the outcome of the SA-JA interaction. During ETI triggered by the *Pst* DC3000 effector avrRpt2, SA levels were greatly increased in infected cells but a crosstalk repression of JA signaling was not detected[46]. It has recently been shown that JA signaling positively regulates RPS2-mediated ETI, whereas the initial activation of the JA response is dependent on SA receptors NPR3 and NPR4 which can promote the degradation of JAZ1 protein with increasing levels of SA in *Arabidopsis*[17]. In poplar, SA and JA interact positively to induce the accumulation of flavonoid phytoalexins to defend against the rust fungus[56]. Therefore, these reports reveal a different interplay between the two plant defense hormones. In the model monocotyledonous plant rice, an antagonistic cross-interaction between JA and SA has not been clearly defined. For example, JA and SA synergistically activated the expression of transcription factor *OsWRKY45*, leading to the upregulation of *PR* genes and enhancing resistance to biotrophs and necrotrophs[18]. However, the detailed interaction between SA and JA in rice immunity has never been investigated and needs to be explored. In this study, we demonstrated that SA and its master regulator OsNPR1 play positive roles in activating JA signaling (Fig. 6). OsNPR1 directly disassociated the OsJAZ9-OsMYC complex (Fig. 5

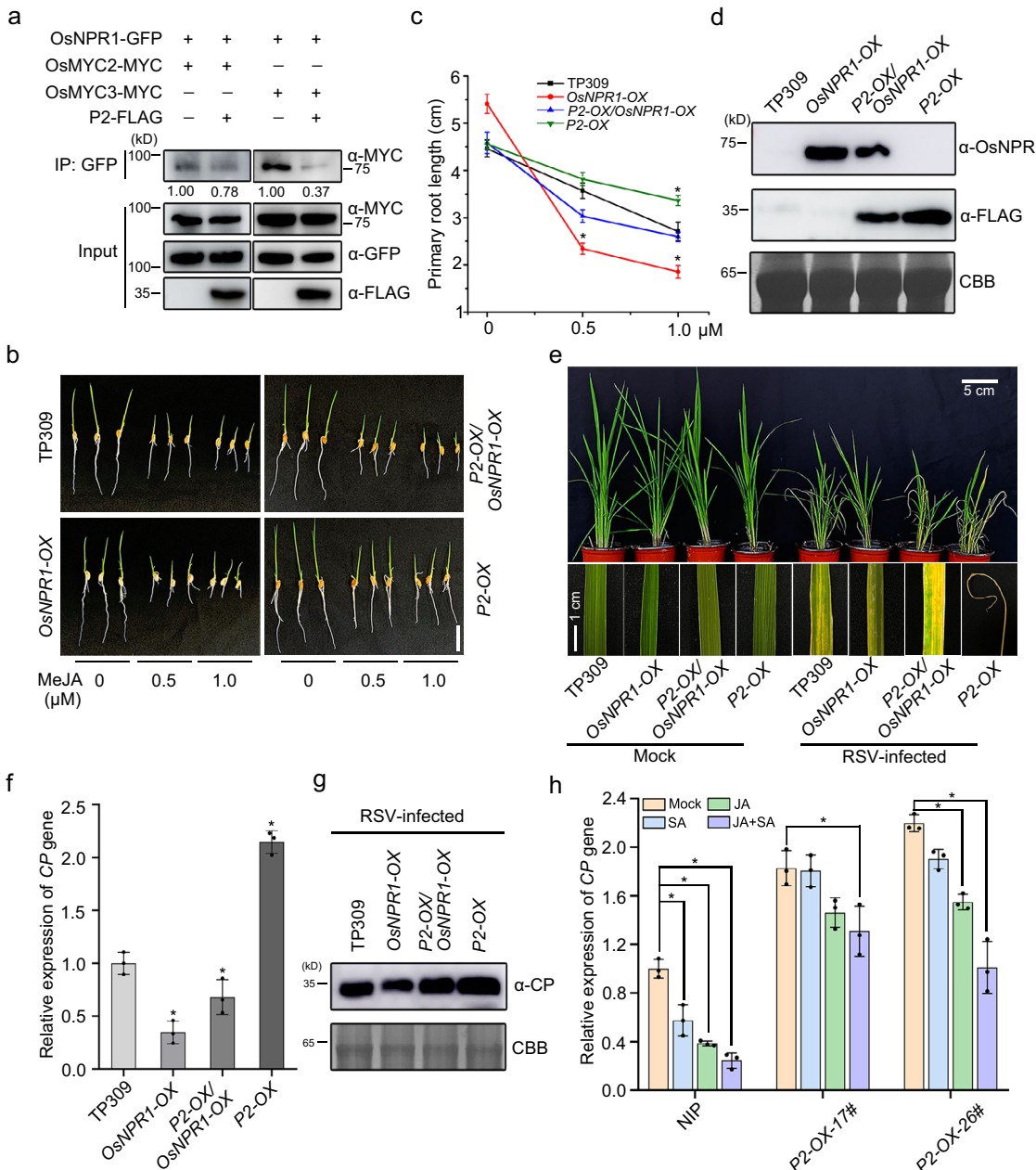

**Fig. 6 | P2 inhibits the interaction between OsNPR1 and OsMYC2/3 to block the response of SA and JA. a** Protein competition analyzed by Co-IP assays *in planta*. OsNPR1-GFP and OsMYC2-MYC or OsMYC3-MYC were infiltrated with or without P2-FLAG in leaves of *N. benthamiana*. The leaves were treated with MG132 (100 μM) or DMSO at 24 hpi and then collected at 24 h for coimmunoprecipitation with GFP beads. **b** Phenotypes of *OsNPR1-OX, OsNPR1-OX/P2-OX, P2-OX* transgenic plants and WT seedlings treated with MeJA. At least 15 germinated seeds were placed in culture solution containing different concentrations of MeJA (0, 0.5 and 1 μM) for about 5 d, scale bar = 5 cm. **c** The primary root lengths of *OsNPR1-OX* (*n* = 15), *OsNPR1-OX/P2-OX* (*n* = 15), *P2-OX* (*n* = 15) transgenic plants and relative to the WT control. Error bars represent SD, values are means ± SD. Significant differences were analyzed using one-way ANOVA followed by Tukey's multiple comparisons test. * at the columns indicate significant differences (*p* ≤ 0.05). **d** The accumulation levels of OsNPR1 proteins in *OsNPR1-OX, OsNPR1-OX/P2-OX, P2-OX* (a transgenic plant expressing P2 with a FLAG tag) transgenic plants and WT analyzed by immunoblot using anti-OsNPR1 and anti-FLAG antibodies. **e** Viral symptoms of *OsNPR1-OX* (*n* = 22), *OsNPR1-OX/P2-OX* (*n* = 25), *P2-OX* (*n* = 25) transgenic plants and WT (*n* = 25)

in response to RSV infection. The phenotypes were observed and photos taken at 30 dpi. Scale bars = 5 cm or 1 cm. **f** Relative mRNA levels of RSV CP detected by RT-qPCR in RSV-infected *OsNPR1-OX, OsNPR1-OX/P2-OX, P2-OX* transgenic plants and WT rice plants. Error bars represent SD, values are means ± SD (*n* = 3 biologically independent replicates per genotype). Significant differences were analyzed using one-way ANOVA followed by Tukey's multiple comparisons test. * at the columns indicate significant differences (*p* ≤ 0.05). **g** The accumulation of RSV CP protein in RSV-infected *OsNPR1-OX, OsNPR1-OX/P2-OX, P2-OX* transgenic plants and WT rice plants determined by western blotting. CBB serves as the loading control to monitor input protein amounts. **h** RSV CP levels measured by RT-qPCR showing the effect of RSV P2 protein on SA- and JA- induced resistance to RSV infection. Error bars represent SD, values are means ± SD (*n* = 3 biologically independent replicates per genotype). Significant differences were analyzed using one-way ANOVA followed by Tukey's multiple comparisons test. * at the columns indicate significant differences (*p* ≤ 0.05). Experiments in (**a**), (**d**) and (**g**) were repeated three times with the similar results. Source data including uncropped scans of gels (**a**, **d** and **g**) and *p* values of statistic tests (**c**, **f** and **h**) are provided in the Source data file.

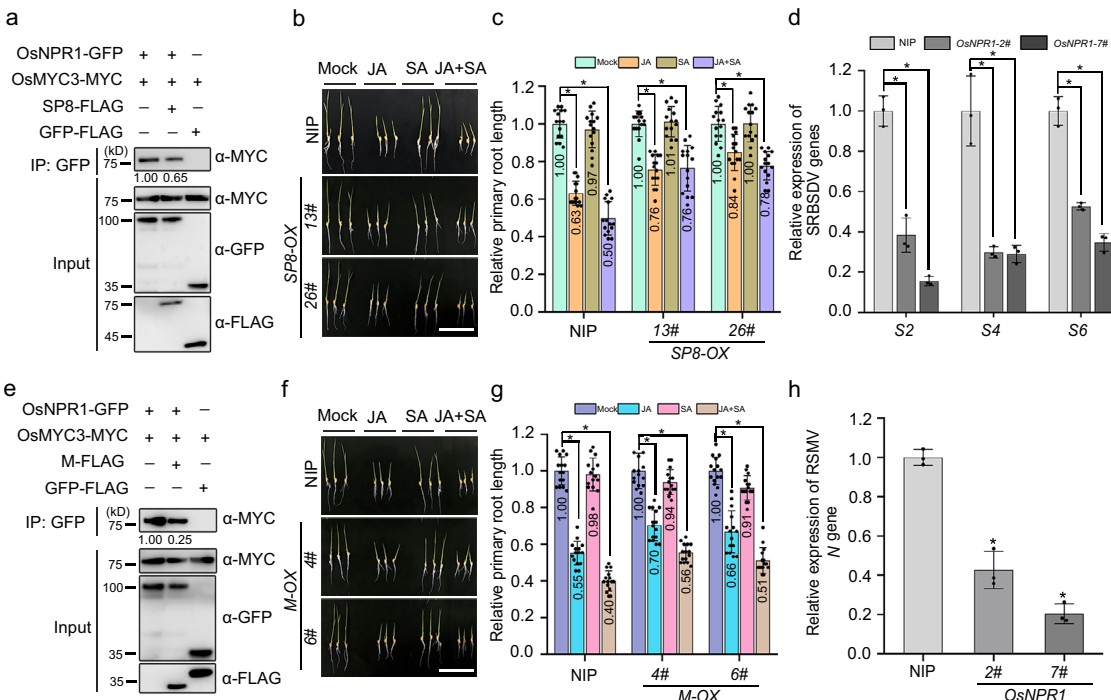

**Fig. 7 | SRBSDV SP8 and RSMV M proteins inhibit the interaction between OsNPR1 and OsMYC3. a**, **e** Protein competition analyzed by Co-IP assays *in planta*. OsNPR1-GFP and OsMYC2-MYC or OsMYC3-MYC were infiltrated with or without SP8-FLAG or M-FLAG in leaves of *N. benthamiana*. The leaves were treated with MG132 (100 μM) or DMSO at 24 hpi and then collected after 24 h for coimmunoprecipitation with GFP beads. **b**, **f** Phenotypes of NIP, SP8-*OX-13#* and SP8-*OX-26#* (b) and NIP, *M-OX-4#* and *M-OX-6#* (f) seedlings treated with SA or/and MeJA. The germinated seeds were planted in concentrations of SA (1 μM) or/and MeJA MeJA (0.5 μM) containing culture solution for about 5 days, scale bar = 5 cm. **c**, **g** The primary root lengths of NIP (*n* = 15), SP8-*OX-13#* (*n* = 15) and SP8-*OX-26#* (*n* = 15) (c) and NIP (*n* = 15), *M-OX-4#* (*n* = 15) and *M-OX-6#* (*n* = 15) (g) seedlings relative to the control. Error bars represent SD, values are means ± SD. Significant differences were analyzed using one-way ANOVA followed by Tukey's multiple comparisons test. * at the columns indicate significant differences (p ≤ 0.05). **d** RT-qPCR results

showing the relative expression levels of SRBSDV (S4, S6, and S10) in SRBSDV-infected OsNPR1 overexpressing lines (named *OsNPR1-2#* and *OsNPR1-7#*) plants compared with SRBSDV-infected control plants. Error bars represent SD, values are means ± SD (*n* = 3 biologically independent replicates per genotype). Significant differences were analyzed using one-way ANOVA followed by Tukey's multiple comparisons test. * at the columns indicate significant differences (*p* ≤ 0.05). **h** RT-qPCR results indicating the relative expression levels of the RSMV *N* gene in RSMV-infected *OsNPR1* overexpressing lines compared with RSMV-infected WT plants. Error bars represent SD, values are means ± SD (*n* = 3 biologically independent replicates per genotype). Significant differences were analyzed using one-way ANOVA followed by Tukey's multiple comparisons test. * at the columns indicate significant differences (p ≤ 0.05). Experiments in (**a**) and (**e**) were repeated three times with the similar results. Source data including uncropped scans of gels (**a** and **e**) and *p* values of statistic tests (**c**, **d**, **g** and **h**) are provided in the Source data file.

and Supplementary Fig. 9) to enhance the transcriptional activation activity of OsMYC2 and then modulated rice antiviral immunity (Figs. 6 and 7). Rice seedling roots were more sensitive to a mixture of SA and JA treatment than to individual treatment with either SA or JA, and this increased sensitivity was dependent on OsNPR1 and OsMYC2/3 transcription factors (Supplementary Fig. 12c–f). In addition, we found that OsNPR1-mediated antiviral defense mainly depended on OsMYC2/3 (Fig. 5h–j). Taken together, these findings showed that OsNPR1 is necessary to SA-mediated activation of the JA signaling pathway. As the key component of JA signaling, MED25 (Mediator complex subunit 25) interacts with MYC2/3 to form a transcription-activating complex[57]. It has been recently reported that NPR1 interferes with the interaction between MYC2 and MED25 to inhibit MYC2/MED25-dependent transcriptional activity in *Arabidopsis*[58]. We first performed yeast two-hybrid and Co-IP assays which showed that OsNPR1 did not interact with OsMED25 (Supplementary Fig. 15a and b). To further investigate whether OsNPR1 influences the association of OsMYC2 with OsMED25, a protein competition Co-IP assays were conducted. Interestingly, we found that OsNPR1 did not affect the OsMYC2-OsMED25 interaction (Supplementary Fig. 15c), but increased the transcriptional activity of OsMYC2 by dissociating the OsJAZ-OsMYC2 complex (Fig. 5c–g). Our results highlight the striking contrast between the SA-JA crosstalk in monocotyledonous rice and that in the dicotyledonous *Arabidopsis*.

There is much evidence that NPR1 is an important modulator of SA and plays a central role in plant immunity[34]. In rice, five NPR1-like proteins have been identified. In contrast to the contrary roles of NPR1 and NPR3/4 in regulating SA signaling and plant defense in *Arabidopsis*[59], OsNPR1 and OsNPR3 were both reported to be positively involved in rice disease resistance[42,60–62]. Rice plants overexpressing OsNPR1 had enhanced resistance to the bacterial pathogen *Xoo* and the blast fungus *Magnaporthe oryzae*[42,63]. Likewise, OsNPR3 is reported to positively regulate the *xa5*-mediated resistance to a bacterial pathogen[64]. Thus, NPRs play different roles in disease resistance depending on the type of host. However, little is known about the function of OsNPR1 protein in resistance to rice viruses. In this study, we found that overexpression of OsNPR1 increased rice resistance to RSV, whereas mutant plants with decreased expression of OsNPR1 were more sensitive to RSV infection than WT plants (Fig. 4). In particularly, we also discovered that OsNPR1 is involved in resistance to different types of rice virus, including the dsRNA virus SRBSDV and the negative ssRNA virus RSMV. Together, our results provide clear evidence that OsNPR1 provides broad-spectrum antiviral resistance in monocotyledonous rice.

Although OsNPR1 positively regulates rice resistance against different types of RNA viruses, it was not sufficient to prevent virus infection entirely, suggesting that RSV, SRBSDV or RSMV may have evolved a strategy to inhibit the OsNPR1-mediated immunity response.

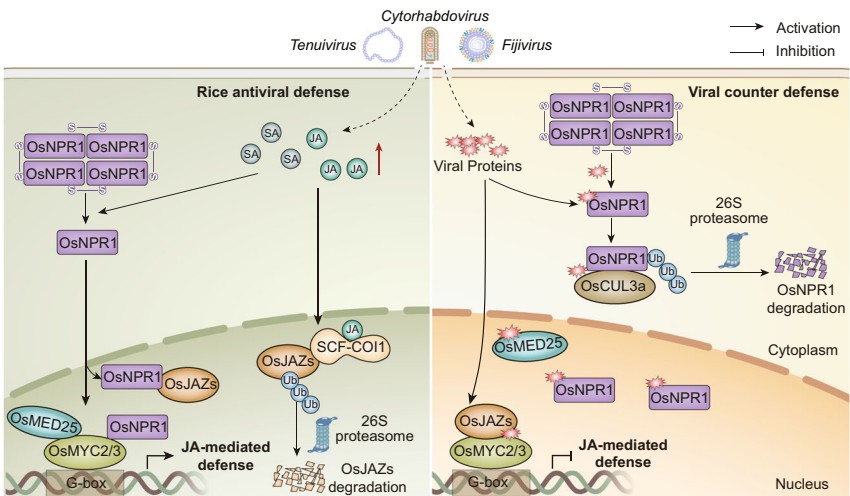

**Fig. 8 | A model showing how the distinct viral proteins suppress OsNPR1-mediated antiviral immunity in rice.** Left panel: following viral infection, JA and SA levels increase in rice plants[23,26,28]. JA binds to and promotes the degradation of OsJAZs, and thus releases OsMYC2/3. Meanwhile, an increase in SA leads to the disassociation of OsNPR1 oligomers into monomers to enter the nucleus. OsNPR1 transcriptionally activates JA signaling by destroying the OsJAZ-OsMYC complex to enhance host antiviral immunity. Right panel: to counteract the host antiviral immunity, different types of viral proteins compete with OsMED25 for binding with OsMYC2, to weaken the interaction of OsMYC2 and OsMED25[8]. Simultaneously, these viral proteins directly target OsNPR1 to inhibit OsNPR1-mediated SA-JA crosstalk, leading to the cooperatively attenuation of the JA response, thus subverting the rice antiviral immune responses. Together, our results reveal that different rice viruses generally to regulate key components of SA and JA signaling to overcome host defense, thereby facilitating viral pathogenicity.

NPR1 is the master regulator of plant defense and its activity is closely restricted by post-translational degradation. In *Arabidopsis* two NPR1 paralogs, NPR3 and NPR4, function as adapters of the Cullin 3 ubiquitin E3 ligase (CUL3) and are involved in the SA-mediated degradation of NPR1[35]. In addition, SUMO3 (small ubiquitin-like modifier 3) dynamically regulates turnip mosaic virus (TuMV) infection and plant immunity through sumoylation of NIb and NPR1[65]. Recent studies indicated that, SA not only induces NPR1 to accumulate in the cytoplasm but also promotes the formation of SA-induced NPR1 condensates (SINCs) to ubiquitinate SINC-localized proteins, suggesting that NPR1 might act as an E3 ligase adapter involved in protein homeostasis[43]. In the absence of pathogen infection, NPR1 disappears from the nucleus via the 26 S proteasome pathway[36]. Upon pathogen infection, the induction of SA causes the oligomeric NPR1 to dissociate into monomers, and the bacterial type III effector AvrPtoB directly targets NPR1 and mediates the degradation of NPR1 through the 26 S proteasome to disrupt plant immunity[37]. Although these few studies show that bacterial or fungal effectors can directly regulate NPR1, it has never been shown to be a target of a viral effector. As a counter-defensive strategy, our findings showed that RSV P2 protein firstly interfered with OsNPR1 oligomerization and then impaired the association of BTB and CTD domains to promote the interaction between OsNPR1 and OsCUL3a to accelerate the degradation of OsNPR1 (Figs. 2 and 3). It is a novel finding that the destabilization and degradation of OsNPR1 by P2 is independent of SA. In addition, we assessed the levels of OsNPR1 in *P2-OX* transgenic and NIP plants after RSV infection. The results showed that the transcriptional and protein levels of OsNPR1 were notably induced in *P2-OX* transgenic and NIP plants after viral infection, while the amounts of OsNPR1 protein induced were less in *P2-OX* than NIP in response to RSV infection (Supplementary Fig. 16). These results further support the conclusion that viral protein P2 promotes the degradation of OsNPR1. Together, these complex molecular interactions demonstrate how viral effectors have evolved to target the key hubs in the plant defense network.

Recent research in our laboratory has shown that several different plant RNA viruses manipulate important components of the JA pathway and result in the transcriptional reprogramming of the JA signaling cascade[8]. In addition, these independently evolved viral proteins target auxin signaling auxin response factor OsARF17 to inhibit its antiviral response[9]. Given that plants have evolved critical defensive hormonal pathways, it is likely that viral effectors including RSV P2, SRBSDV P8 and RSMV M, manipulate these hormone pathways and provide an effective strategy that may help many viruses to infect. In this study, we found that expression of RSV P2 in *OsNPR1-OX* transgenic plants resulted in increased accumulation of RSV and symptoms (Fig. 6e, f, g). These results further illustrate that RSV P2 suppresses the OsNPR1-mediated JA response. In a JA sensitivity assay the synergistic effects of JA-SA in decreasing root length were strongly reduced in plants overexpressing SRBSDV SP8 or RSMV M (Fig. 7c, g). The endogenous JA and SA concentrations in transgenic plants expressing viral proteins (*P2-OX*, *SP8-OX* and *M-OX*) were decreased compared with control NIP plants (Supplementary Fig. 17). These results reveal that viral proteins can directly manipulate the key components of JA and SA signaling to overcome host defense. In conclusion, our results reveal that OsNPR1 transcriptionally activates JA signaling by dissociating the OsJAZ-OsMYC complex to enhance host antiviral immunity. To counteract the host antiviral defense, different types of viral proteins compete with OsMED25 for binding with OsMYC2, to weaken OsMYC2-mediated JA response[8]. Meanwhile, viral proteins directly target OsNPR1 to inhibit OsNPR1-mediated SA-JA crosstalk, leading to the cooperatively attenuation of the JA response, thus subverting the rice antiviral immune responses (Fig. 8). Collectively, our data reveal a novel mechanism that a variety of viral proteins widely interfere with the interaction between OsNPR1 and OsMYC2/3 to obstruct the OsNPR1-mediated activation of JA signaling and thus facilitate viral infection.

## Methods
### Plant materials and growth conditions
Seeds of wildtype rice varieties used in this study were NIP, TP309 and Xiushui 11 (XS11). The transgenic rice plants expressing SP8 and M were in NIP background[8,10]. The *OsNPR1-OX* transgenic rice plants were in the TP309 background[42]. The OsNPR1 overexpression transgenic rice plants were created in a NIP background. The *npr1-cas-1#* and *npr1-cas-3#* mutants were in a XS11 background[66]. The OsMYC2 and OsMYC3 mutant *Ri-m2m3* lines were constructed in this study. The relative

expression levels of genes in transgenic rice plants were detected by RT-qPCR and western blotting assays. RSV-infected plants were kindly provided by Professor Tong Zhou (Jiangsu Academy of Agricultural Sciences, China). Isolates of SRBSDV and RSMV were kindly provided by Professor Guohui Zhou (South China Agricultural University, China). RSV-, SRBSDV- and RSMV- infected plants were maintained in our laboratory. The seeds were geminated first and grown into the greenhouse maintained at 28-30 °C with a 14/10 h light/dark cycle. The *N. benthamiana* plants used in follow-up expression assays were grown in black plastic bowls at 25 °C and a 14/10 h photoperiod prior for two weeks.

## Insect vectors and virus infection

RSV was transmitted by *Laodelphax striatellus* (small brown planthopper, SBPH) and SRBSDV by *Sogatella furcifera* (white-blacked planthopper, WBPH)[9,67]. To acquire RSV or SRBSDV, 2-3 virus-free nymphs of SBPH or WBPH were fed on RSV/SRBSDV-infected rice plants for 3-5 days. The source RSV/SRBSDV-infected rice plants were kept in our laboratory. Then the insects were removed onto healthy Wuyujing3 rice seedlings for about 10 days to accomplish viral circulation in the planthoppers. Afterwards, SBPH carrying RSV or WBPH carrying SRBSDV were transferred to transgenic rice plants at the 3 to 4-leaf stage for 3 days. For RSMV inoculation experiments, we used 2-3 virus-free nymphs of leafhoppers to feed on RSMV-infected rice plants for 3-5 days. Virus-infected leafhoppers were then placed on 3 to 4-leaf stage seedlings for 3 days. At the same time, the rice seedlings were infested with the same number of virus-free insects as a negative control. After the feeding period, the insects were removed completely. The inoculated plants were grown in a growth room to observe symptoms. Infection of RSV, SRBSDV or RSMV in these inoculated plants was confirmed by RT-PCR at 30 dpi. The symptoms of RSV-infected plants were graded based on the severity of symptoms on new young leaves and photographs of 4 to 5 plants with representative symptoms were taken. The specific primers used to test for viral infection are listed in Supplementary Table 1.

## Vector construction and plant transformation

To generate transgenic plants, the coding sequences (CDS) of RSV P2, SRBSDV P8 and RSMV M were cloned into the pCAMBIA1300 vector, driven by the CaMV 35 S promoter with FLAG tag. To produce rice OsMYC2 and OsMYC3 RNAi lines, a highly conserved region between OsMYC2 (Os10g0575000) and OsMYC3 (Os01g0705700) was amplified and inserted into the pTCK303 RNAi vector to generate *Ri-m2m3* transgenic rice plants. To generate *osnpr1-cas* knockout mutants, the target sequence of *OsNPR1* gene was introduced into pLYsgRNA-OsU6b to produce sgRNA. The sgRNA expression cassette was then introduced into pYLCRISPR/Cas9Pubi-H vector[66]. The rice transformation was done by BioRun (Wuhan, China). For Y2H assays, the full-length CDS of OsNPR1 and its truncated variants containing BTB, ANK and CTD were amplified using the specific primers listed in Supplementary Table 1. The PCR products were cloned into the yeast bait vector pGBKT7. The full-length CDS of RSV P2, SRBSDV P8 and RSMV M were cloned into the yeast prey vector pGADT7.

For LCI assays, the full-length CDS of OsCUL3a, the full-length CDS of OsNPR1 and their truncated mutants including BTB and CTD were ligated into pCAMBIA1300-nLuc or pCAMBIA1300-cLuc vectors, respectively.

For BiFC assays, the full-length CDS of OsNPR1, OsMYC2, OsMYC3, OsJAZ5, OsJAZ9 and RSV P2 were amplified by PCR with specific primers and then individually introduced into the N-terminus of YFP or the C-terminus of YFP vectors.

For Co-IP assays, the binary vector pCAMBIA1300 was used to generate various expression vectors. Briefly, the CDS of OsCUL3a, OsMYC2, OsMYC3, OsJAZ9, OsMED25 and the truncated mutations of OsNPR1 (BTB, CTD and △CTD in which the CTD domain was deleted from OsNPR1) were inserted into the pCAMBIA1300vector, driven by the CaMV 35 S promoter with MYC, FLAG, and GFP tag, respectively.

For in vitro pull-down, the full-length sequences of RSV P2, OsNPR1 and OsCUL3a were amplified by PCR with specific primers and then constructed individually into the pGEX6P1 vector and pET28a vectors, respectively, to express GST-tagged or MBP-HIS-tagged fusion proteins.

For luciferase assays, the promoter regions of *OsMADS1* and *OsNOMT* were respectively inserted into the pGreenII0800-Luc vector (*pMADS1* and *pNOMT*) as reporters. The Renilla LUC (REN) gene of pGREENII0800-LUC was used as a control.

All the primers used are listed in Supplementary Table 1.

## Bimolecular fluorescence complementation (BiFC) assays

For BiFC assays, the different vectors used were as follows: OsNPR1-cYFP, nYFP-OsNPR1, P2-cYFP, nYFP-P2, OsMYC2-cYFP, nYFP-OsMYC2, OsMYC3-cYFP, nYFP-OsMYC3, OsJAZ9-cYFP, nYFP-OsJAZ9. These different constructs were transformed into *Agrobacterium tumefaciens* strain GV3101 by electroporation. Then different vector combinations were infiltrated into *N. benthamiana* leaves for 48 h and the YFP fluorescence was observed using a Leica TCS SP10 confocal laser microscopy. To capture YFP signals, the 514 nm excitation laser wavelength was used. Three biological repeats were conducted for all experiments.

## Yeast two-hybrid assays (Y2H)

The recombinant plasmids with various interaction pairs were co-transformed into the yeast strain AH109 as described in the manufacturer's instructions with minor modification (Clontech, CA, USA). The transformants were grown on a culture medium lacking Leu and Trp (SD/-L-T) for 3 days at 30 °C, and then the positive colonies were selected and transferred to SD/-Leu/-Trp/-His/-Ade selection plates for 3 days at 30 °C. The yeast growth was observed and photographed for the interaction test. The AD empty vector was co-transformed with the BD expression vector to test for gene auto-activation. The experiments were repeated at least three times with similar results.

## Co-Immunoprecipitation (Co-IP) assays

For Co-IP assays, the different combinations were co-expressed in *N. benthamiana* leaves. After infiltration for 24 h, MG132 (Sigma) was added at a final concentration of 100 μM to prevent RSV P2-induced degradation of OsNPR1. After infiltration for 40-48 h, the tobacco leaf tissues were harvested and ground into powder with liquid nitrogen and used for Co-IP assays. The native total protein was extracted using IP lysis buffer (Thermo Scientific, Cat. no. 87788) with 10 mM DTT, 1× EDTA-free protease inhibitor cocktail (Roche, Basel, Switzerland)[9]. After incubation for 30 min at 4 °C, the homogenate was centrifuged at 12000 g, 4 °C for 10 min, and the process repeated. The supernatant was incubated with 20 μL Pierce™ anti-c-Myc magnetic beads (Thermo Scientific, USA), anti-FLAG M2 beads (Sigma-Aldrich, USA), anti-GFP-trap beads (Chromotek, Germany) and Protein A/G OsMYC2 antibody beads, respectively, for approximately 2 h at 4 °C with gentle shaking. Protein beads were prewashed three times with 1×PBS. After co-incubation, the immunoprecipitates were washed three times with 1×PBS and resuspended in 50-100 μL 2×SDS-PAGE sample buffer containing 500 mM Tris-HCl, 50% glycerin, 10% SDS, 1% bromophenol blue and 2% β-mercaptoethanol, pH=6.8). Subsequently, the proteins were analyzed by anti-FLAG (1:5000 dilution, Cat#HT201-01, TransGen) / MYC (1:5000 dilution, Cat#HT101-01, TransGen) / GFP (1:5000 dilution, Cat#HT801-01, Genscript) / OsNPR1 (1:2000 dilution, our lab) / OsMYC2 (1:2000 dilution, our lab) / OsJAZ9 (1:2000 dilution, our lab) antibody.

## Plant protein extraction and western blotting

The denatured proteins of rice or tobacco leaf tissues were extracted with SDS lysis buffer containing 100 mM Tris-HCl, pH 6.8, 10% SDS and 0.5 M DTT[25]. For nonreduced conditions, protein was extracted with IP lysis buffer without DTT and denatured with 5×SDS-PAGE sample buffer lacking DTT[37]. Polyclonal antibody of RSV CP was used to test for RSV. We used ABclonal Biotechnology (Wuhan, China) company to generate the rabbit polyclonal antibodies against OsNPR1 (1:2000 dilution, our lab). After incubating with IgG-HRP anti-body, the protein membranes were imaged using ECL substrate and photographed by the BIO-RAD ChemiDoc MP Imaging System. Total proteins were stained with Coomassie brilliant blue (CBB) to confirm equal loading. Protein levels were quantified using ImageJ software (https://imagej.nih.gov/ij/).

## In vitro pull-down assays

The vectors pGEX6P1-P2, pGEX6P1-OsNPR1, OsNPR1-MBP-HIS, P2-MBP-HIS and OsCul3a-MBP-HIS were transferred into *Escherichia coli* strain Rosetta (DE3) (WEIDI Biotech, Shanghai, China) and the empty pGEX6P1, MBP-HIS were used as the negative controls. The bacteria were cultivated at 28 °C with shaking at 200 rpm. To induce protein expression, a final concentration of 0.8 mM IPTG was added to the cultures when the $OD_{600}$ was 0.6 to 0.8 and incubated at 28 °C for 5-7 h for GST-P2, GST-OsNPR1, MBP-HIS-P2, MBP-HIS-OsNPR1 and MBP-HIS-OsCUL3a. For competitive pull-down assays, 5 μg MBP-HIS-OsNPR1 was mixed with 5 μg GST-OsNPR1 and then mixed with 10 μg MBP-HIS (negative control), 1 μg MBP-HIS-P2 or 10 μg MBP-HIS-P2, respectively, for each experimental group. These mixed proteins were incubated and retained on glutathione GST sepharose beads at 4 °C for 2 h. The beads were washed three times with 1×PBS. The proteins immobilized on beads were separated by SDS-PAGE and visualized by anti-MBP (1:5000 dilution, Cat#HT701-01, Genscript) / GST (1:5000 dilution, Cat#A00130, Genscript) antibody. The total proteins were stained with Coomassie Brilliant Blue (CBB). For interaction assays, 5 μg MBP-HIS-OsCUL3a was mixed with 5 μg GST-OsNPR1 for each experimental group, and then 1 or 10 μg MBP-HIS-P2 was added for incubation. 5 μg GST was the negative control. These mixed proteins were then incubated and immobilized onto glutathione GST sepharose beads at 4 °C for 2 h. The beads were washed three times with 1×PBS. The proteins immobilized on beads were separated by SDS-PAGE and visualized by anti-HIS (1:3000 dilution, Cat#ab18184, abcam) / GST (1:5000 dilution, Cat#A00130, Genscript) / MBP (1:5000 dilution, Cat#HT701-01, Genscript) antibody.

## Protein degradation assays

For the cell-free protein degradation assay, 10-day-old transgenic rice seedlings overexpressing OsNPR1 were harvested and ground into a fine power in liquid nitrogen. Total protein was extracted by degradation buffer containing 25 mM Tris-HCl pH 7.5, 10 mM NaCl, 10 mM $MgCl_2$, 5 mM DTT and 10 mM ATP. Equal amounts of extracts were added to GST or GST-P2 purified proteins at 37 °C for each experimental group. After incubation, the samples were collected at 0, 3, 6 and 8 h for western blotting assays to detect the expression of OsNPR1 using anti-OsNPR1. The proteins were stained with Coomassie Brilliant Blue (CBB). For ubiquitination assays, leaves of rice and *N. benthamiana* plants were ground into powder and extracted with IP lysis buffer containing 100 μM MG132 and 10 mM DTT. Then, the crude extracts were immuno-precipitated using anti-GFP-trap beads and Protein A/G-Magnetic beads. The bound proteins were analyzed using anti-Ubiquitin (1:2000 dilution, P4D1-sc-8017, AbcamSanta Cruz Biotechnology) and anti-OsNPR1 antibodies (1:2000 dilution, our lab).

## Cell fractionation assays

Nuclear-cytoplasmic fractionation assays were performed as the following descriptions. Rice leaves were collected and ground in liquid nitrogen, and then extracted in a lysis buffer (20 mM Tris-HCl, pH 7.5, 20 mM KCl, 2 mM EDTA, 2.5 mM $MgCl_2$, 25% glycerol, 250 mM sucrose) with addition of 100 μM MG132, a protease inhibitor cocktail (Roche, Basel, Switzerland), 5 mM DTT and 1 mM PMSF. The crude extracts were filtered through a double layer of Miracloth and centrifuged at 10,000 g for 10 min at 4 °C. The supernatant was collected as the cytoplasmic fraction. The precipitate was washed four times with NRBT buffer (20 mM Tris-HCl, pH=7.4, 25% glycerol, 2.5 mM $MgCl_2$, and 0.2% Triton X-100) and then resuspended in 500 μL of NRB2 buffer (20 mM Tris-HCl, pH=7.5, 250 mM sucrose, 10 mM $MgCl_2$, 0.5% Triton X-100) supplemented with protease inhibitor cocktail and 100 μM MG132. The samples were carefully overlaid on top of 500 μL NRB3 buffer (20 mM Tris-HCl, pH=7.5, 1.7 M sucrose, 10 mM $MgCl_2$, 0.5% Triton X-100) with added protease inhibitor cocktail. Subsequently, the samples were centrifuged at 16,000 g for 45 min at 4 °C. The pellet was resuspended in 500 μL lysis buffer as the nuclear fraction[68]. The proteins were separated by SDS-PAGE for immunoblot analyses. Actin (1:5000 dilution, ab-mart, Cat#M20009) and histone H3 (1:5000 dilution, Huabio, Cat#ET1701-64) proteins were used as the respective cytoplasmic and nuclear markers.

## Firefly luciferase (LUC) complementation imaging (LCI) assays

LCI assays were conducted as described with minor modifications. The constructs were transformed into *A. tumefaciens* strain GV3101 using electroporation. The vectors were infiltrated and expressed in the combinations described in *N. benthamiana* leaves with a final concentration of $OD_{600}$ 1.0 for approximately 48 h. Subsequently, 0.2 mM luciferin substrate (Perkin Elmer, EU) was diluted with 0.01% Triton X-100 and infiltrated into leaves under darkness for 5 min and the luciferase activity was tested using a low-light cooled CCD imaging apparatus (NightOWL II LB983). In the competition LCI assays, the GV3101 strains harboring NLUC-OsNPR1 and CLUC-OsNPR1 with P2-FLAG were co-infiltrated into *N. benthamiana* leaves, GUS-MYC was co-infiltrated as a negative control. At least three biological repeats were conducted for all experiments.

## Dual-luciferase transient transcriptional activity assays (Dual-LUC)

For luciferase assays, the constructs of effectors and reporters were transformed into *Agrobacterium tumefaciens* strain GV3101. The combinations of effectors and reporters were then transformed into *N. benthamiana* leaves with a final concentration of $OD_{600}$ 1.0 for 48 h. For interference in dual-LUC assays, the OsMYC2-MYC and OsNPR1-MYC were used as effectors, the promoter regions of *OsMADS1* and *OsNOMT* were ligated into the pGreenII0800-Luc vector (*pMADS1* and *pNOMT*) as reporters. The dual-LUC assays were performed using the Luciferase Reporter Assay System (Promega, Madison, WI) following the manufacturer's instructions. Briefly, two disks with a diameter of 0.6 cm were collected from the leaves using a puncher for shattering by an automatic oscillator, then 300 μl 1×Passive Lysis Buffer (PLB) was added to promote rapid lysis. The tissue supernatant was collected by centrifuging at 12000 g for 30 s at 4 °C. 20 μl aliquots of each lysate were transferred into wells of a 96-well plate. The relative luciferase activity was analyzed using LUC/REN ratios. The REN luminescence was considered as an internal control. At least three biological repeats were conducted for all experiments.

## Total RNA extraction and quantitative RT-PCR

Total RNA was isolated from rice or tobacco leaves using TRIzol reagent (Invitrogen, Carlsbad, CA, USA). Complementary DNA was synthesized using the fast quant RT kit (Vazyme, Nanjing, China). The resulting cDNA was used as the template for RT-PCR and RT-qPCR. RT-qPCR was conducted using the ChamQTM SYBR qPCR Master Mix (Low ROX Premixed) and ABI7900HT Sequence Detection System (Applied Biosystems, Carlsbad, CA, USA). The rice actin gene OsUBQ5

(AK061988) was used as an internal reference. The results were analyzed by the $2^{-\Delta\Delta Ct}$ method and shown as means ± SD (n = 3). At least three biological repeats were conducted for all experiments. The RT-qPCR primer sequences used are listed in Supplementary Table 1.

## Chromatin immunoprecipitation (ChIP)-qPCR

The EpiQuikTM Plant ChIP Kit (Epigentek, Brooklyn, USA) was used for ChIP-qPCR assays, following the manufacturer's instructions. Briefly, the two-week-old rice seedlings of NIP, *OsNPR1-7#* and *Ri-m2m3* plants were harvested for total DNA extraction. The seedlings were cut into 2 mm strips and cross-linked with 1% formaldehyde in fixation solution using vacuum aspiration. Subsequently, 0.125 M glycine solution was added to quench the reaction under vacuum for 5 min. The samples were washed with double-distilled water and stored at −80 °C for use. The chromatin was sheared into 200-1000 bp fragments by ultrasonic disruption. The fragmented chromatin solution was immunoprecipitated by anti-OsNPR1 or anti-OsMYC2 polyclonal antibody (1:2000 dilution, produced in our lab) bound Protein A/G-Magnetic beads for 2 h. Negative control samples were prepared using immunoglobulin G (IgG). The enriched DNA fragments and input control were purified following the manufacturer's instructions, and the immunoprecipitated DNA was used for qPCR. The relative enrichment was calculated as a percentage of the input. At least three biological repeats were conducted for all experiments. The primers are listed in Supplementary Table 1.

## Analysis of SA and JA concentrations

Rice leaves of transgenic *P2-OX*, SP8-*OX, M-OX* and NIP (control) were collected at the 3 to 4-leaf stage and powdered in liquid nitrogen, respectively. Samples (about 0.2 g of leaf powder) were extracted in 2 mL of 80% methanol solution (containing 0.1% formic acid) and purified by an Oasis mode anion exchange (MAX) solid phase extraction (SPE) column. JA and SA were extracted and analyzed followed by ultra-high-performance liquid chromatography-triple quadrupole mass spectrometry (UPLC-MS/MS) 23, 28. The UPLC system included an Acquity UPLC™ System (Waters) quaternary pump coupled to an autosampler and a ACQUITY UPLC HSS T3 column (100 × 2.1 mm, 1.7 μm). For analyses of JA and SA using the ESI (Electron Spray Ionization, Xevo TQ-S) source in negative ion mode, with 2.5 kV capillary voltage, 18 V cone voltage, and 25 or 30 collision voltage, respectively. Quantitative data were processed by the Masslynx V4.1 software. Differences between samples were analyzed using one-way ANOVA with Tukey's least significant difference tests. The sample was replicated three times, each of which consisted of at least 4-5 pooled plants.

## JA or SA sensitivity analysis

For JA sensitivity assays (primary root inhibition assays) seeds of transgenic and control rice plants were germinated and then transferred into rice nutrient solution wells of a 96-well plate with 0.1 or 0.5 μM MeJA under 8 h light/16 h dark, at 30 °C for 5 days. The primary root lengths were measured and photos taken to evaluate the sensitivity of the JA response. For SA sensitivity assays, all transgenic and control rice plants were prepared as described above. The germinated seeds were transferred into rice nutrient solution wells of a 96-well plate containing different concentrations of SA (0, 3, 5, 10 or 20 μM) under 8 h light/16 h dark, at 30 °C for 5 days. The primary root lengths were measured and the phenotypes were photographed. For JA and SA mixed treatment, all transgenic and control rice plants were prepared as described above. The germinated seeds were transferred into rice nutrient solution wells of a 96-well plate containing 0.1 μM MeJA and 3 μM SA under 8 h light/16 h dark, at 30 °C for 5 days. The primary root lengths were measured and the phenotypes were photographed. At least 15 transgenic seedlings were used for each line.

## Hormone treatments of RSV-inoculated rice seedlings

Stock solutions of SA (Cat#S607-25G, Sigma) and MeJA (Cat#M1068, TCI) in 100% ethanol were diluted with sterile distilled water containing 0.1% Triton X-100. RSV-inoculated rice seedlings were sprayed with SA (500 μM) or/and JA (100 μM) and 1% Triton X-100 as control. Each treatment used at least 20-30 seedlings. The symptoms of RSV-infected plants were observed at 30 dpi.

## Statistical analysis

Statistical significance analysis, quantitative real-time PCR analysis, and dual-luciferase reporter system were analyzed using one-way ANOVA with Tukey's least significant difference tests. Each experiment was repeated at least three times, and data are represented as the mean. * at the columns indicate significant differences ($p \le 0.05$). All analyses were performed using ORIGIN 8.0 software. For immunoblot quantification analysis, the intensities of bands were quantified with Image J.

## Reporting summary

Further information on research design is available in the Nature Portfolio Reporting Summary linked to this article.

## Data availability

Sequence data from this article can be found in the rice genome annotation project database under the following accession numbers: OsNPR1; Os01g09800; OsCUL3a; Os02g51180; OsMYC2; Os10g42430; OsMYC3, Os01g50940; OsJAZ1, Os04g55920; OsJAZ3, Os08g33160; OsJAZ4, Os09g23660; OsJAZ5, Os04g32480; OsJAZ6, Os03g28940; OsJAZ7, Os07g42370; OsJAZ8, Os09g26780; OsJAZ9, Os03g08310; OsJAZ10, Os03g08330; OsJAZ11, Os03g08320; OsJAZ12, Os10g25290; OsJAZ13, Os10g25230; OsJAZ14, Os10g25250; OsJAZ15, Os03g27900; OsMADS1, Os03g11614; OsNOMT, Os12g13800; OsMED25; Os09g13610; OsUBQ5, Os10g39620. The authors declare that all raw data supporting the findings of this study can be found within the paper and its Supplementary Files. Source data are provided with this paper.

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

## Acknowledgements

We thank Prof. Zuhua He (Shanghai Institutes for Biological Sciences, Chinese Academy of Sciences) and Prof. Ran Li (Zhejiang University) for providing OsNPR1 transgenic plant seeds, Prof. Jianxiang Wu (Zhejiang University) for providing viral protein antibodies and Prof. Tong Zhang (South China Agricultural University) for providing RSMV-infected plants. We thank Prof. Zhengqing Fu (University of South Carolina) for their important suggestions and Prof. Mike Adams for critically reading and improving the manuscript. This work was funded by the National Key Research and Development Plan of China (2021YFD1400500), National Natural Science Foundation of China (32272555, 32270149, 32022072, 32001888), Young Elite Scientist Sponsorship Program by CAST (YESS20210121), Zhejiang Provincial Natural Science Foundation (LZ22C140001, LQ21C140005) and Ningbo Science and Technology Innovation 2025 Major Project (2019B10004).

## Author contributions

H.Z., J.C., and Z.S. designed the experiments; H.Z., F.W., W.S., and Z.S. performed the experiments; H.Z., F.W., W.S., Z.Y., L.L., Q.M., X.T., Z.W., Y.L., J.L., F.Y., and Z.S. analyzed the data; H.Z. and Z.S. wrote the manuscript; H.Z., J.C., and Z.S. revised the manuscript; All authors discussed the results and commented on the manuscript.

## Competing interests

The authors declare no competing interests.

## Additional information

**Peer review information** : *Nature Communications* thanks Kranthi Mandadi, Zhonglin Mou and the other, anonymous, reviewer(s) for their contribution to the peer review of this work. A peer review file is available.

