## [Peer Review File · Nature Communications]

Different viral effectors suppress the hormone-mediated antiviral immunity of rice coordinated by OsNPR1Reviewer #1 (Remarks to the Author):

In this manuscript, the authors first found that the Rice stripe virus (RSV) P2 protein interacts with OsNPR1 and then conducted a series of experiments to demonstrate that P2 interacts with the CTD of OsNPR1, which allows the BTB domain of OsNPR1 to interact with OsCul3, leading to 26S proteasome-mediated degradation. OsNPR1 interacts with several OsJAZ proteins and OsMYC2/3 to enhance OsMYC2/3 transcriptional activity. P2-promoted degradation of OsNPR1 attenuates the OsMYC2/3 signaling, resulting in increased susceptibility to RSV.

This manuscript provides an interesting model to explain how the RSV P2 protein suppresses rice immunity. However, the conclusion was built mainly upon transgenic OsNPR1 overexpression lines. I would like the authors to consider the following concerns and comments.

Major concerns

1. The authors claim throughout the manuscript that SA and OsNPR1 trigger/activate JA signaling. This claim is misleading. It seems that root inhibition is an output of both SA and JA signaling in rice. SA strongly inhibits primary root growth in an OsNPR1- and OsMYC2/3-dependent manner, indicating that SA-OsNPR1/OsMYC2/3-root inhibition is an SA signaling pathway. Here OsMYC2/3 are SA signaling components. On the other hand, JA appears to be mainly dependent on OsMYC2/3 (partially on OsNPR1). The synergism between SA and JA on root growth relies on both OsNPR1 and OsMYC2/3. It is possible that OsNPR1 can increase the sensitivity of OsMYC2/3 to JA, but SA by itself can activate the OsNPR1/OsMYC2/3 module. It would be interesting to know if SA can inhibit primary root length in the absence of JA and vice versa.

2. Overexpression of OsNPR1 does not equal to SA-triggered signaling. Although the authors tested the effect of RSV P2 on the resistance provided by overexpression of OsNPR1, its effect on SA- and JA-triggered resistance in TP309 (the wild type) is not shown. I guess that both SA and JA can induce resistance to RSV in TP309. What is the effect of RSV P2 on this resistance? The biological relevance of RSV P2-induced OsNPR1 degradation should also be investigated in TP309, not just in OsNPR1 overexpression lines. Note that the basal OsNPR1 level is very low (Fig. 6d). How about OsNPR1 levels in TP309 after RSV infection? It is important to look at what happens to OsNPR1 in TP309-RSV interaction.

3. If RSV P2 is an effector protein, loss of P2 would reduce the virulence of RSV. The authors may have shown this in their previous publications. If that is the case, please mention this result in the introduction and add the reference. By the way, some information about the P2 protein in the introduction would be helpful.

4. I am not sure if it is reliable to solely use overexpression of P2 to study its function. Is there a way to detect the RSV P2 protein or to tag the P2 protein in RSV, so we can follow the P2 protein levels during RSV infection? I wonder if P2 levels can ever reach to the levels in the P2-OX transgenic plants.

Minor comments

1. Fig 1b, the GFP-FLAG protein amount used was much less than that of P2-FLAG, which may contribute to the negative result (no pulling down of OsNPR1-MYC).

2. Fig 2f legend, should it be "Protein A/G OsNPR1 antibody beads"? Also, although OsNPR1 levels are lower in P2-OX/OsNPR1-OX, the amounts of OsNPR1 precipitated by the antibody beads are similar. Did you adjust the loading or add MG132?

3. Line 62, that, not "than". Line 69, the reference should be #53.

4. Line 516 to 517, compared to OsNPR1-OX, expression of RSV P2 in OsNPR1-OX transgenic plants resulted in INCREASED RSV accumulation and symptoms.

5. Line 519, Fig. 6f, g. There is no Fig. 6i.

Reviewer #2 (Remarks to the Author):

The article by Zhang et al describes that RSV P2 protein physically interacts with OsNPR1 and interferes with OsNPR1 oligomerization. P2 protein accelerates OsNPR1 degradation by the ubiquitin-proteasome system. P2 protein interacts with CTD domain of OsNPR1, which facilitates the interaction between OsCUL3a and BTB domain of OsNPR1 which normally interacts with CTD domain to block the interaction between OsNPR1 CTD domain and OsCUL3a. The authors show that OsNPR1 interacts with OsMYC2/3 and OsJAZ protein that regulate JA-responsive gene expressions. The authors find that OsNPR1 enhances OsMYC2 activity transcribing JA-regulated gene, OsMADS1, suggesting that OsNPR1 has a positive role in JA signaling pathway. It is supported by the result that OsNPR1 regulates JA+SA and JA-mediated inhibition of root growth. The authors suggest that P2 alleviates OsNPR1-mediated JA response, root growth inhibition, through inhibition of the interaction between OsNPR1 and OsMYC2/3 protein. Furthermore, this article shows that OsNPR1 has an important role in antiviral defense against three distinct viruses, RSV, SRBSDV and RSMV. P2 expression enhance susceptibility against RSV, suggesting that P2-mediated OsNPR1 degradation leads to the enhanced susceptibility through suppression of OsNPR1-mediated SA-JA synergistic antiviral defense. Finally, the authors show that OsNPR1 overexpression confers enhanced resistance against SRBSDV and RSMV, and that both SRBSDV SP8 and RSMV M proteins interfere with the interaction between OsNPR1 and OsMYC3, suggesting a general role of OsNPR1 in SA-JA-mediated antiviral defense and viral counter strategy focusing on OsNPR1-OsMYC2/3 interaction in rice.

Although NPR1-MYC interaction in JA-regulated genes are already reported in Arabidopsis (Nomoto et al., 2021, Cell Reports, 37, 110125), this article contains novel findings and valuable information on SA-JA crosstalk especially in antiviral defense. However, I observed several points that need to be addressed and clarified.

Major points

- SA-JA synergy mediated by OsNPR1 and OsMYC2/3 is a novel finding. However, the authors indicated only a single example, OsMADS1 gene. You should show additional evidence showing SA-JA synergism in the other JA-regulated genes.
- The authors show that OsNPR1 synergistically activates OsMYC2-regulated JA-regulated gene, OsMADS1. Additional data on how OsNPR1 activates JA-regulated gene expression in concert with OsMYC2 is required to draw the authors' conclusion. Does OsNPR1 affect OsMYC2 - OsMED25 interaction?
- Does P2 expression affect OsJAZ-OsMYC interaction? It is important to understand the mechanism to suppress JA signaling in virus infection.
- Does OsNPR1 protein level actually decrease in RSV-infected rice cells?

Minor points

Line 531: P2 > M

Line 536: Please describe the strain/isolate names of viruses (RSV, SRBSDV, RSMV) used in this study

Line 552: RSMV is maintained in RSMV-infected rice plants?

Line 618: delete PCR

Line 639: RBSDV-infected rice > rice

Discussion: Recent report in Arabidopsis shows that AtNPR1 is recruited to JA-responsive promoter, and that AtNPR1 physically interacts with AtMYC2 and inhibits the transcriptional activation by interfering the interaction between AtMYC2 and AtMED25. Please discuss the difference between rice and Arabidopsis.

Reviewer #3 (Remarks to the Author):

In this study by Zhang et al., the function of a viral protein (P2) of the Rice stripe virus (RSV) was dissected. It interacts with OsNPR1 and promoted its degradation by enhancing the association of OsNPR1 and OsCUL3a. The study also explored SA-JA crosstalk and how P2 can influence JA-mediated responses via SA. In addition to RSV, the authors also described how other viral proteins from SRBSDV and RSMV also interact with NPR1 and influence SA-JA crosstalk.

Interaction of P2, NPR1, Cullin3, and NPR1 turnover-

Authors employed a variety of complementary biochemical approaches to test their hypotheses and each experiment is technically sound with appropriate controls and replicated. The results and conclusions surrounding RSV P2 interaction with OsNPR1 leading to its turnover/degradation are well done and justified based on the experimental results (Fig. 1, 2, 3). A few suggestions in this section are below:

Recently Arabidopsis NPR1 was shown to interact with Cullin3 in so-called condensates (SINC) of various cellular proteins and target them for a turnover during plant immune response (Zavaliev et al., 2020, Cell 182(5):1093-1108). In this context, where is the OsNPR1 turnover happening during RSV infection? Cytosol or nucleus? In the presented model (Fig. 7i), it is shown as a nuclear event. Is there any evidence? It is important to know this spatial context, especially considering the NPR1 SINCS. Also, based on the in vitro IP assays (Fig. 2), P2 disrupts the oligomerization of NPR1, which primarily occurs in the cytosol. So, it is likely the turnover could be happening in the cytosol, and it is possible the P2 could be targeted to the NPR1 SINCS during infection perhaps to suppress NPR1 function.

Characterization of OsNPR1 OE and npr1-cas lines-

The authors suggest that OsNPR1 OE lines are resistant to RSV, while npr1-cas edited lines are more susceptible than WT. However, looking at the phenotypes of the infected plants at 20 dpi (Fig. 4a, 4e, 6e), all the transgenic lines appear more or less like the controls. There is some change in stature, but that is variable. See 4d vs. 6e for OsNPR1OE lines. There are no clear differences.

To support the visual phenotypic data, the authors presented the disease symptoms as quantitative graphs (Fig. 4b, 4f, 6f % disease symptoms). The disease index showed differences among the lines regarding the different grades of disease symptoms on the leaves (H, I, II, and III as classified in Fig. S6). It appears that ~20-30 plants were used for these experiments. Several things need to be clarified here. Was the grading done by counting the number of leaves per plant with the different symptoms, or is it done on a whole plant level? Based on the images of the infected plants, each infected plant seems to have a mixture of different symptoms. How did the authors account for these admixtures of symptomatic leaves in each plant? Also, the data/disease indexes were not subjected to appropriate statistical tests. Overall, this grading scheme appears very subjective, and not clear how it was done.

To support the disease phenotype/index data, the authors also presented effects on RSV levels by measuring RSV CP transcript and protein by RT-PCR and immunoblotting. There appear to be expected differences in RSV CP levels among the controls and NPR1 OE, npr1-cas lines (Fig. 4c, d, g, h). However, the levels of RSV CP are inconsistent and do not correlate to transcript levels. For instance, in Fig. 4d, the control TP309 has a good amount of RSV CP, while in 4h the control XS11 has barely any detectable RSV CP, despite both showing comparable disease symptoms (Fig. 4a, e). Also, why are the levels of RSV CP transcript in npr1-cas lines #1 and #3 inversely proportional to their protein levels? Which leaf grades (I, II, III) were sampled for these RNA and protein assays? These variations could have stemmed from a single sampling time point late in the infection. The sampling was done at 20 dpi after RSV inoculation (or 30 dpi as per methodology?). At this late stage, when disease symptoms have quite manifested, often the levels of the virus do not correlate with disease symptoms since necrotic tissues (although resulting from severe virus infection) may end up not having as much virus since it's a dying tissue/cell. Other nutrient/biochemical limitations could also complicate the amount of virus the cells can support depending on the tissue that was collected (newly emerging vs. mature leaves). Hence, comparing such virus accumulation as a time course or at an earlier stage would be more meaningful and

appropriate.

P2, NPR1, the SA-JA crosstalk.

The conclusions in the subsequent sections related to P2 (and other viral proteins) and OsNPR1 in SA-JA crosstalk (Fig. 5) are not well-justified and are frankly oversimplified. It is known that SA and JA are key hormones involved in diverse plant growth and developmental responses. A complex interplay of interactions exists among these pathways. The authors showed that RSV P2 also interacted with OsMYC2/3 and disrupted the interactions of NPR1 with MYC2/3. In a series of experiments, they conclude that the P2, and other unrelated viral proteins, interact and disturb the NPR1-mediated JA signaling to promote virus infection. The broad conclusion on the effects on virulence/virus infection is not supported by the data, since these are derived from experiments of a root inhibition assay used as a read-out for SA-JA signaling. No SA-JA-mediated defense responses were assayed, nor effects on virus/virulence were assessed in that context. Generalizing the conclusion to more viruses/viral proteins cannot be also justified until these shortcomings are addressed by appropriate experiments. A few are suggested below:

The SA-JA crosstalk is rather complex and deserves to be carefully assessed by conducting thorough genetic/epistasis experiments with various SA-JA genes/mutants and viral proteins. The phenotypes of SA/JA mutants in the context of virus infection could offer clues to whether there was a significant perturbation of JA defenses leading to resistance or susceptibility outcomes.

Furthermore, the determination of SA, and JA hormone levels, and effects on downstream transcriptional targets of the SA-JA mediated signaling pathways would provide a better understanding of the role of these viral proteins to perturb SA-JA mediated defenses.

Minor comments:

Choice of words: Replace "obvious" in several places with other choices (e.g., noticeable, discernable).

To maintain the flow and the focus of the study on antiviral defenses/responses, the entire section on page 10 could be omitted. "OsNPR1 triggered JA signaling by disturbing the OsJAZ-OsMYC2 complex and activating the transcriptional activity of OsMYC2." This section describes the native interactions of OsNPR1 with JA components in the rice SA-JA crosstalk with no bearing on antiviral responses. In fact, it Deserves to be presented as a separate study with further analysis of the observed SA-JA synergism in rice.

Response to Reviewers' Comments

We are very grateful to the reviewers for their constructive and valuable suggestions on the manuscript. We have carefully taken these into consideration in preparing this revision and have performed the additional experiments suggested. We believe that the revised manuscript has addressed the reviewers' concerns and has resulted in a better paper. Comments received are shown in black below, with our response in red font. The following are our point-by-point answers to the reviewers' questions and comments.

Reviewer #1 (Remarks to the Author):

In this manuscript, the authors first found that the Rice stripe virus (RSV) P2 protein interacts with OsNPR1 and then conducted a series of experiments to demonstrate that P2 interacts with the CTD of OsNPR1, which allows the BTB domain of OsNPR1 to interact with OsCul3, leading to 26S proteasome-mediated degradation. OsNPR1 interacts with several OsJAZ proteins and OsMYC2/3 to enhance OsMYC2/3 transcriptional activity. P2-promoted degradation of OsNPR1 attenuates the OsMYC2/3 signaling, resulting in increased susceptibility to RSV.

This manuscript provides an interesting model to explain how the RSV P2 protein suppresses rice immunity. However, the conclusion was built mainly upon transgenic OsNPR1 overexpression lines. I would like the authors to consider the following concerns and comments.

Response: Thank you very much for your helpful and supportive suggestions. We have added the detailed information in the revised manuscript as suggested by the Reviewer #1 (see below).

Major concerns

1.The authors claim throughout the manuscript that SA and OsNPR1

trigger/activate JA signaling. This claim is misleading. It seems that root inhibition is an output of both SA and JA signaling in rice. SA strongly inhibits primary root growth in an OsNPR1- and OsMYC2/3-dependent manner, indicating that SA-OsNPR1/OsMYC2/3-root inhibition is an SA signaling pathway. Here OsMYC2/3 are SA signaling components. On the other hand, JA appears to mainly dependent on OsMYC2/3 (partially on OsNPR1). The synergism between SA and JA on root growth relies on both OsNPR1 and OsMYC2/3. It is possible that OsNPR1 can increase the sensitivity of OsMYC2/3 to JA, but SA by itself can activate the OsNPR1/OsMYC2/3 module. It would be interesting to know if SA can inhibit primary root length in the absence of JA and vice versa.

Response: Thanks for these comments. JA treatment, but not SA treatment, has been widely reported to inhibit root growth in plants, and this inhibitory effect is enhanced in plants when JA signaling is activated^{1, 2, 3}. Consistent with previous reports, we found that the root length of rice seedlings was efficiently inhibited by 0.5 μ M MeJA treatment but not by SA (1 μ M) treatment (Supplementary Fig. 10). We then chose a mixture of 0.5 μ M MeJA and 1 μ M SA to treat the rice seedlings, and found that root length was greatly inhibited by the mixture compared with either of the single treatments. This showed that SA enhances the JA-mediated root inhibitory effect. In further experiments, this synergistic effect of SA on JA-inhibition of root length was significantly attenuated in the OsNPR1 mutant *npr1-cas* and the OsMYC2/3 mutants *Ri-m2m3* (Supplementary Fig. 10). These results indicated that the enhancement of JA signaling by SA was largely dependent on OsNPR1 and OsMYC2/3 (See below, Figure 1).

Figure 1. The effect of SA and JA on the primary root length of OsNPR1 mutants and *Ri-m2m3* transgenic plants.

References:

1. Li, L. et al. A class of independently evolved transcriptional repressors in plant RNA viruses facilitates viral infection and vector feeding. *Proc. Natl. Acad. Sci. USA* **118**, e2016673118 (2021).
2. Acosta, L., Gasperini, D., Chételat, A., Stolz, S., Santuar, L. & Farmer, E. Role of NINJA in root jasmonate signaling. *Proc. Natl. Acad. Sci. USA*. **110**, 15473-15478 (2013).

3. Yang, Z., He, C., Ma, Y., Herde, M., & Ding, Z. Jasmonic acid enhances AI-induced root growth inhibition. *Plant Physiol.* **173**, 1420-1433 (2017).

2. Overexpression of OsNPR1 does not equal to SA-triggered signaling. Although the authors tested the effect of RSV P2 on the resistance provided by overexpression of OsNPR1, its effect on SA- and JA-triggered resistance in TP309 (the wild type) is not shown. I guess that both SA and JA can induce resistance to RSV in TP309. What is the effect of RSV P2 on this resistance?

Response: We thank the reviewer for this suggestion. In the revised manuscript, we have added RSV inoculation experiments under JA/SA treatments in WT (NIP) and P2-OX transgenic plants. As shown in Figure 6h, The mixture of SA and JA treatment greatly increased rice resistance to RSV infection compared with either SA or JA alone in wildtype plants. However, this synergistic effect of SA and JA on plant defense was significantly alleviated in P2-OX transgenic plants (Figure 6h). These results indicated that the SA-JA synergistic antiviral defense was obstructed by viral effector P2 protein (Line 406-416) (See below Figure 2).

Figure 2. The effect of RSV P2 protein on SA- and JA- induced resistance to RSV infection.

The biological relevance of RSV P2-induced OsNPR1 degradation should also be investigated in TP309, not just in OsNPR1 overexpression lines. Note that the basal OsNPR1 level is very low (Fig. 6d). How about OsNPR1 levels in TP309 after RSV infection? It is important to look at what happens to OsNPR1 in TP309-RSV interaction.

Response: We thank the reviewer for this suggestion. In our revised manuscript, we have added experimental data to further confirm P2-induced OsNPR1 protein degradation in wildtype NIP and *P2-OX* transgenic plants (Supplemental Fig. 3c and d, Line 185-188) (See below Figure 3).

Figure 3. The protein and mRNA levels of OsNPR1 in *P2-OX* transgenic and NIP rice plants.

In addition, as suggested by the reviewer, we have added data on the level of OsNPR1 in *P2-OX* transgenic and NIP plants after RSV infection by RT-qPCR and western blotting assays. The transcriptional level of *OsNPR1* was significantly induced in both *P2-OX* transgenic and NIP plants after inoculation with RSV, but the increased amount of OsNPR1 protein was significantly less in *P2-OX* transgenic plants than in NIP (Supplemental Fig. 15). These results suggested that OsNPR1 was transcriptionally induced in response to RSV infection, while the accumulation of OsNPR1 protein was

blocked in the presence of viral protein P2, further supporting our conclusion that P2 protein promotes the degradation of OsNPR1 in rice (Line 545-551) (See below, Figure 4).

Figure 4. The effect of P2 on OsNPR1 protein and mRNA levels.

3. If RSV P2 is an effector protein, loss of P2 would reduce the virulence of RSV. The authors may have shown this in their previous publications. If that is the case, please mention this result in the introduction and add the reference. By the way, some information about the P2 protein in the introduction would be helpful.

Response: We have added this information in the revised manuscript (Line 56-63).

4. I am not sure if it is reliable to solely using overexpression of P2 to study its function. Is there a way to detect the RSV P2 protein or to tag the P2 protein in RSV, so we can follow the P2 protein levels during RSV infection? I wonder if P2 levels can ever reach to the levels in the P2-OX transgenic plants.

Response: Unfortunately, our attempts to produce an effective P2 antibody were not successful. Usually, viral proteins were extensively expressed due to virus continuous replication and proliferation in host cell. To address the

reviewer's concern, we detected the transcriptional levels of *P2* in RSV-infected plants and *P2-OX* plants, and found that greater amounts of *P2* were expressed in RSV-infected plants than in *P2-OX* (see Figure 5, below).

Figure 5. The levels of *P2* in RSV-infected rice and *P2* transgenic rice plants.

Minor comments

1. Fig 1b, the GFP-FLAG protein amount used was much less than that of *P2*-FLAG, which may contribute to the negative result (no pulling down of OsNPR1-MYC).

Response: Thanks. We have re-done the Co-IP assays to confirm the interaction between *P2* and OsNPR1 proteins in *N. benthamiana* leaves (Fig. 1b in revised manuscript and below Figure 6).

Figure 6. Co-IP assays showing that OsNPR1 protein interacted with P2 protein in *Nicotiana benthamiana* leaves.

2. Fig 2f legend, should it be “Protein A/G OsNPR1 antibody beads”? Also, although OsNPR1 levels are lower in P2-OX/OsNPR1-OX, the amounts of OsNPR1 precipitated by the antibody beads are similar. Did you adjust the loading or add MG132?

Response: Thanks for this comment. We have corrected this description in Fig 2f legend. For better analysis of OsNPR1 ubiquitination levels, we have used similar amounts of the precipitated OsNPR1 protein and added MG132 to the protein extraction buffer. We have added this information to the Fig 2f legend (Line 1159-1163).

3. Line 62, that, not “than”. Line 69, the reference should be #53.

Response: We have corrected these mistakes carefully in our revised manuscript (Line 65).

4. Line 516 to 517, compared to OsNPR1-OX, expression of RSV P2 in OsNPR1-OX transgenic plants resulted in increased RSV accumulation and

symptoms.

Response: Done (Line 400-405).

5. Line 519, Fig. 6f, g. There is no Fig. 6i.

Response: Done (Line 404).

Reviewer #2 (Remarks to the Author):

The article by Zhang et al describes that RSV P2 protein physically interacts with OsNPR1 and interferes with OsNPR1 oligomerization. P2 protein accelerates OsNPR1 degradation by the ubiquitin-proteasome system. P2 protein interacts with CTD domain of OsNPR1, which facilitates the interaction between OsCUL3a and BTB domain of OsNPR1 which normally interacts with CTD domain to block the interaction between OsNPR1 CTD domain and OsCUL3a. The authors show that OsNPR1 interacts with OsMYC2/3 and OsJAZ protein that regulate JA-responsive gene expressions. The authors find that OsNPR1 enhances OsMYC2 activity transcribing JA-regulated gene, OsMADS1, suggesting that OsNPR1 has a positive role in JA signaling pathway. It is supported by the result that OsNPR1 regulates JA+SA and JA-mediated inhibition of root growth. The authors suggest that P2 alleviates OsNPR1-mediated JA response, root growth inhibition, through inhibition of the interaction between OsNPR1 and OsMYC2/3 protein.

Furthermore, this article shows that OsNPR1 has an important role in antiviral defense against three distinct viruses, RSV, SRBSDV and RSMV. P2 expression enhance susceptibility against RSV, suggesting that P2-mediated OsNPR1 degradation leads to the enhanced susceptibility through suppression of OsNPR1-mediated SA-JA synergistic antiviral defense. Finally, the authors show that OsNPR1 overexpression confers enhanced resistance against SRBSDV and RSMV, and that both SRBSDV SP8 and RSMV M proteins interfere with the interaction between OsNPR1 and OsMYC3, suggesting a general role of OsNPR1 in SA-JA-mediated antiviral defense

and viral counter strategy focusing on OsNPR1-OsMYC2/3 interaction in rice.

Although NPR1-MYC interaction in JA-regulated genes are already reported in Arabidopsis (Nomoto et al., 2021, Cell Reports, 37, 110125), this article contains novel findings and valuable information on SA-JA crosstalk especially in antiviral defense. However, I observed several points that need to be addressed and clarified.

Response: We are very grateful to Reviewer #2 and have tried to incorporate further experiments in the revised manuscript as suggested (see below).

Major points

- SA-JA synergy mediated by OsNPR1 and OsMYC2/3 is a novel finding. However, the authors indicated only a single example, *OsMADS1* gene. You should show additional evidence showing SA-JA synergism in the other JA-regulated genes. The authors show that OsNPR1 synergistically activates OsMYC2-regulated JA-regulated gene, *OsMADS1*. Additional data on how OsNPR1 activates JA-regulated gene expression in concert with OsMYC2 is required to draw the authors' conclusion.

Response: Thanks for the reviewer's helpful suggestion. Previous work has shown that (in addition to the *OsMADS1* gene) OsMYC2 directly binds to the *OsNOMT* promoter, a key enzyme for JA-induced sakuranetin production^{1, 2}. Thus, we fused the promoter of *OsNOMT* with a firefly luciferase (LUC) to construct the *pOsNOMT::LUC* vector. The results were similar to those with the *OsMADS1* promoter: the transcriptional activation activity of OsMYC2 on the *OsNOMT* promoter was significantly increased in the presence of OsNPR1, further confirming that OsNPR1 enhances the transcriptional activation activity of OsMYC2 (Fig 5 d, Line 330-336) (See below Figure 7).

Figure 7. OsNPR1 elevated the transcriptional activation activity of OsMYC2 to *OsMADS1* and *OsNOMT* genes.

References:

1. Cai, Q. et al. Jasmonic acid regulates spikelet development in rice. *Nat Commun.* 5, 3476 (2014).
2. Ogawa, S. et al. OsMYC2, an essential factor for JA-inductive sakuranetin production in rice, interacts with MYC2-like proteins that enhance its transactivation ability. *Sci Rep* 7, 40175 (2017).

• Does OsNPR1 affect OsMYC2 - OsMED25 interaction?

Response: To address this, we first performed yeast two-hybrid and Co-IP assays that showed that OsNPR1 did not directly interact with OsMED25. In a subsequent protein competition Co-IP assay, OsNPR1 did not affect the OsMYC2-OsMED25 interaction (Supplementary Fig. 14, Line 487-500 and see below Figure 8).

Figure 8. A protein competition Co-IP assay showing that OsNPR1 did not influence the OsMYC2-OsMED25 interaction.

· Does P2 expression affect OsJAZ-OsMYC interaction? It is important to understand the mechanism to suppress JA signaling in virus infection.

Response: We have previously shown that P2 protein did not disturb the association of OsJAZ11 and OsMYC2/3, and P2 protein did not alter the stability of OsJAZ proteins¹.

References:

1. Li, L. et al. A class of independently evolved transcriptional repressors in plant RNA viruses facilitates viral infection and vector feeding. *Proc. Natl. Acad. Sci. USA* **118**, e2016673118 (2021).

Does OsNPR1 protein level actually decrease in RSV-infected rice cells?

Response: please see our response to reviewer #1 (above Figure 4). Briefly, in response to RSV infection, the plant immune pathway including JA and SA pathway were activated^{1,2}. Consistently, the transcriptional and protein

levels of OsNPR1 were significantly induced after RSV infection (Figure 4). However, to counter host antiviral defense, RSV encodes viral effector P2 to target and promote the degradation of OsNPR1 for facilitating its infection. As shown in our results (Supplementary Fig. 15, Line 545-551) , the increased extent of OsNPR1 protein in response to virus infection was significantly weakened in *P2-OX* transgenic plants compared to wild-type NIP. Overall, our results reveal a novel mechanism in the co-evolutionary arms race between plant defense and viral counter defense.

References:

1. Wang, Q. et al. STV11 encodes a sulphotransferase and confers durable resistance to rice stripe virus. *Nat Commun.* **5**, 4768 (2014).
2. Yang, Z. et al. Jasmonate signaling enhances RNA silencing and antiviral defense in rice. *Cell Host Microbe.* **28**, 89-103 (2020).

Minor points

Line 531: P2 > M

Response: done (Line 580).

Line 536: Please describe the strain/isolate names of viruses (RSV, SRBSDV, RSMV) used in this study

Response: done (Line 587-591).

Line 552: RSMV is maintained in RSMV-infected rice plants?

Response: Yes. We have added a detailed description in the revised manuscript (Line 606-608).

Line 618: delete PCR

Response: done.

Line 639: RBSDV-infected rice > rice

Response: done.

Discussion: Recent report in *Arabidopsis* shows that AtNPR1 is recruited to JA-responsive promoter, and that AtNPR1 physically interacts with AtMYC2 and inhibits the transcriptional activation by interfering the interaction between AtMYC2 and AtMED25. Please discuss the difference between rice and *Arabidopsis*.

Response: We thank the reviewer for this comment. We have done this in the revised manuscript (Line 487-500).

Reviewer #3 (Remarks to the Author):

In this study by Zhang et al., the function of a viral protein (P2) of the Rice stripe virus (RSV) was dissected. It interacts with OsNPR1 and promoted its degradation by enhancing the association of OsNPR1 and OsCUL3a. The study also explored SA-JA crosstalk and how P2 can influence JA-mediated responses via SA. In addition to RSV, the authors also described how other viral proteins from SRBSDV and RSMV also interact with NPR1 and influence SA-JA crosstalk.

Interaction of P2, NPR1, Cullin3, and NPR1 turnover-

Authors employed a variety of complementary biochemical approaches to test their hypotheses and each experiment is technically sound with appropriate controls and replicated. The results and conclusions surrounding RSV P2 interaction with OsNPR1 leading to its turnover/degradation are well done and justified based on the experimental results (Fig. 1, 2, 3). A few suggestions in this section are below:

Response: We thank the reviewer #3 for evaluation of our manuscript and provide valuable suggestions. In response to the concern of the reviewer, we have added following experiments in the revised manuscript as suggested (see below).

Recently *Arabidopsis* NPR1 was shown to interact with Cullin3 in so-called

condensates (SINC) of various cellular proteins and target them for a turnover during plant immune response (Zavaliev et al., 2020, Cell 182(5):1093-1108). In this context, where is the OsNPR1 turnover happening during RSV infection? Cytosol or nucleus? In the presented model (Fig. 7i), it is shown as a nuclear event. Is there any evidence? It is important to know this spatial context, especially considering the NPR1 SINC. Also, based on the in vitro IP assays (Fig. 2), P2 disrupts the oligomerization of NPR1, which primarily occurs in the cytosol. So, it is likely the turnover could be happening in the cytosol, and it is possible the P2 could be targeted to the NPR1 SINC during infection perhaps to suppress NPR1 function.

Response: We are grateful for the reviewer's helpful comments. To clarify where viral protein promotes the degradation of OsNPR1, we performed western blotting analysis using the nuclear and cytoplasmic fractions of *OsNPR1-OX* and *OsNPR1-OX/P2-OX* samples (Supplementary Fig. 4b). The results indicated that P2 significantly decreased the level of OsNPR1 in the cytoplasm, whereas only a small reduction was found in the nucleus. We further tested the ubiquitination levels of OsNPR1 in the different fractions (Supplementary Fig. 4c, Line 201-210). There was significantly increased poly-ubiquitination of OsNPR1 in the cytoplasm, but not in the nucleus, in the presence of viral protein P2 (see below Figure 9). Thus, the degradation of OsNPR1 promoted by P2 seems to occur mainly in the cytoplasm. In the revised manuscript, we have corrected the model (Fig. 8). The focus of this study was how viral proteins impede JA-SA crosstalk to facilitate viral infection. Whether P2 protein can target NPR1 SINC to regulate protein homeostasis is worthy of future investigation and we are grateful for the valuable suggestions.

Figure 9. Nuclear-cytoplasmic fractionation analysis the influence of P2 on OsNPR1 ubiquitination.

Characterization of OsNPR1 OE and npr1-cas lines-

The authors suggest that OsNPR1 OE lines are resistant to RSV, while npr1-cas edited lines are more susceptible than WT. However, looking at the phenotypes of the infected plants at 20 dpi (Fig. 4a, 4e, 6e), all the transgenic lines appear more or less like the controls. There is some change in stature, but that is variable. See 4d vs. 6e for OsNPR1OE lines. There are no clear differences.

Response: We apologize that our description caused some confusion. To address the reviewer's concern, we re-performed the RSV inoculation experiments. In Fig. 4a, 4e and 6e (lower panel), we have added enlarged detail images. Upon challenging with RSV, the areas of typical yellow stripes and curling or death of the young leaves represent the degree of disease symptoms. Based on the severity of symptoms on leaves, we clearly observed that *OsNPR1-OX* plants had milder virus symptoms with discontinuous yellow

stripes and necrotic streaks on the leaves (Fig. 4a and 6e), while *npr1-cas* plants had severe curling or death of the young leaves (Fig. 4e, and see below Figure 10).

Figure 10. *OsNPR1* confers resistance to RSV infection in rice.

To support the visual phenotypic data, the authors presented the disease symptoms as quantitative graphs (Fig. 4b, 4f, 6f % disease symptoms). The disease index showed differences among the lines regarding the different grades of disease symptoms on the leaves (H, I, II, and III as classified in Fig. S6). It appears that ~20-30 plants were used for these experiments. Several things need to be clarified here. Was the grading done by counting the number of leaves per plant with the different symptoms, or is it done on a whole plant level? Based on the images of the infected plants, each infected plant seems to have a mixture of different symptoms. How did the authors account for these admixtures of symptomatic leaves in each plant? Also, the data/disease

indexes were not subjected to appropriate statistical tests. Overall, this grading scheme appears very subjective, and not clear how it was done.

Response: In the revised manuscript, we have added a more detailed description of the method (Line 612-617). For RSV inoculation assays, each independent experiment contains 20-30 seedlings. The inoculated plants were monitored for viral symptoms at about 20 to 30 dpi. The symptoms of RSV-infected plants were graded based on the severity of symptoms on **new young leaves**, since this is where viral systemic infection mainly exists. The photos of 4 to 5 plants with mixed symptoms were taken. In addition, we re-conducted a statistical analysis of the disease index (See Figure 11).

Figure 11. A statistical analysis of the disease index in *OsNPR1-OX* and *npr1-cas* mutant plants.

To support the disease phenotype/index data, the authors also presented effects on RSV levels by measuring RSV CP transcript and protein by RT-PCR and immunoblotting. There appear to be expected differences in RSV CP levels among the controls and NPR1 OE, *npr1-cas* lines (Fig. 4c, d, g, h). However, the levels of RSV CP are inconsistent and do not correlate to transcript levels. For instance, in Fig. 4d, the control TP309 has a good amount

of RSV CP, while in 4h the control XS11 has barely any detectable RSV CP, despite both showing comparable disease symptoms (Fig. 4a, e). Also, why are the levels of RSV CP transcript in npr1-cas lines #1 and #3 inversely proportional to their protein levels? Which leaf grades (I, II, III) were sampled for these RNA and protein assays? These variations could have stemmed from a single sampling time point late in the infection. The sampling was done at 20 dpi after RSV inoculation (or 30 dpi as per methodology?). At this late stage, when disease symptoms have quite manifested, often the levels of the virus do not correlate with disease symptoms since necrotic tissues (although resulting from severe virus infection) may end up not having as much virus since it's a dying tissue/cell. Other nutrient/biochemical limitations could also complicate the amount of virus the cells can support depending on the tissue that was collected (newly emerging vs. mature leaves). Hence, comparing such virus accumulation as a time course or at an earlier stage would be more meaningful and appropriate.

Response: Thanks for the reviewer's helpful comment. Firstly, as for the concern that the protein levels of RSV CP in the control TP309 and XS11 were inconsistent, this is due to different exposure times on different PVDF membranes. To solve the reviewers' concern, we repeated the RSV inoculation experiments and re-detected the levels of RSV CP transcript and protein by RT-qPCR and western blotting (Fig. 4c, 4d, 4g and 4h). In addition, samples taken at different times after RSV inoculation were tested. The new young leaves with disease symptoms were collected and mixed for these RNA and protein assays. Each biological sample with 4 to 5 pooled young leaves of plants was evaluated with two technical replicates. There were usually three biological replicates. Results from RT-qPCR and western blotting were consistent with the symptoms and demonstrate that OsNPR1 plays key roles in rice antiviral defense against RSV infection (See above, Figure 10).

The conclusions in the subsequent sections related to P2 (and other viral

proteins) and OsNPR1 in SA-JA crosstalk (Fig. 5) are not well-justified and are frankly oversimplified. It is known that SA and JA are key hormones involved in diverse plant growth and developmental responses. A complex interplay of interactions exists among these pathways. The authors showed that RSV P2 also interacted with OsMYC2/3 and disrupted the interactions of NPR1 with MYC2/3. In a series of experiments, they conclude that the P2, and other unrelated viral proteins, interact and disturb the NPR1-mediated JA signaling to promote virus infection. The broad conclusion on the effects on virulence/virus infection is not supported by the data, since these are derived from experiments of a root inhibition assay used as a read-out for SA-JA signaling. No SA-JA-mediated defense responses were assayed, nor effects on virus/virulence were assessed in that context. Generalizing the conclusion to more viruses/viral proteins cannot be also justified until these shortcomings are addressed by appropriate experiments. A few are suggested below:

The SA-JA crosstalk is rather complex and deserves to be carefully assessed by conducting thorough genetic/epistasis experiments with various SA-JA genes/mutants and viral proteins. The phenotypes of SA/JA mutants in the context of virus infection could offer clues to whether there was a significant perturbation of JA defenses leading to resistance or susceptibility outcomes.

Response: We thank the reviewer #3 for this helpful suggestions. We realized that the genetic/epistasis experiments with SA-JA mutants would be necessary to support a strong conclusion before manuscript submission, so we had crossed *OsNPR1* overexpression plants with an *OsMYC2/3* RNAi mutant, and obtained the homozygous *OsNPR1-7/Ri-m2m3-6* hybrid plants from the experimental field in this autumn. Therefore, we assessed the sensitivity of *OsNPR1-7/Ri-m2m3-6* hybrid plants to RSV infection and found that similar to *Ri-m2m3*, *OsNPR1-7/Ri-m2m3-6* hybrid plants were hypersensitive to RSV infection compared with wildtype and *OsNPR1* overexpression plants (Fig. 5g, h, i, and Supplementary Fig. 11b, c, Line 374-382). These results suggested that *OsNPR1*-mediated antiviral defense was largely depended on JA

signaling key transcription factors OsMYC2/3.

Figure 11. The sensitivity of *OsNPR1-7/Ri-m2m3-6#* hybrid plants to RSV infection.

Furthermore, the determination of SA, and JA hormone levels, and effects on downstream transcriptional targets of the SA-JA mediated signaling pathways would provide a better understanding of the role of these viral proteins to perturb SA-JA mediated defenses.

Response: We thank the reviewer for pointing out this relevant issue that we had previously overlooked. To confirm whether different viral proteins impact endogenous JA or SA hormone contents, we analyzed JA and SA concentrations in *P2-OX*, *SP8-OX* and *M-OX* transgenic rice plants as Supplementary Fig. 16. The JA and SA contents in these transgenic rice plants were significantly decreased compared with those in control NIP plants. The results further support our conclusion that these unrelated viral proteins could perturb SA-JA mediated antiviral immunity (Line 567-570, and see below Figure 12).

Figure 12. Effect of viral proteins on the JA and SA concentrations in rice leaves.

Minor comments:

Choice of words: Replace “obvious” in several places with other choices (e.g., noticeable, discernable).

Response: done (Line 148, 171, 172, 219, 238, and 345).

To maintain the flow and the focus of the study on antiviral defenses/responses, the entire section on page 10 could be omitted. “OsNPR1 triggered JA signaling by disturbing the OsJAZ-OsMYC2 complex and activating the transcriptional activity of OsMYC2.” This section describes the native interactions of OsNPR1 with JA components in the rice SA-JA crosstalk with no bearing on antiviral responses. In fact, it deserves to be presented as a separate study with further analysis of the observed SA-JA synergism in rice.

Response: We are grateful for the reviewer’s comments. In the revised manuscript, we have reorganized this section, put the results of root sensitivity into Supplementary Fig.10, and added the viral inoculation on the *OsNPR1-7/Ri-m2m3* hybrid plants, which further supports the conclusion that OsNPR1 triggering of antiviral defense largely depends on OsMYC2/3.

Reviewer #1 (Remarks to the Author):

The authors have addresses all of my major and minor concerns. I am satisfied with the authors' answers and changes and have no more questions and comments.

Reviewer #2 (Remarks to the Author):

I thank the authors for their careful revisions. The paper has been significantly improved. The authors have addressed my comments, but two points need to be considered.

1) (Regarding the first major points)

To draw the authors' conclusion on the OsNPR1 function in the transcriptional regulation of JA genes, further evidence is required on how OsNPR1 activates JA-regulated gene expression in concert with OsMYC2. Are there any data on it? Is the OsNPR1/OsMYC2 complex actually recruited in the promoter region of JA-regulated genes such as OsMADS1 and OsNOMT in vivo? and does OsJAZs dissociate from the promoter region by OsNPR1?

2) (Regarding the third major point)

The author mentioned in the rebuttal letter, "We have previously shown that P2 protein did not disturb the association of OsJAZ11 and OsMYC2/3, ...". In this article, P2 accelerated the degradation of OsNPR1 (Fig2), and OsJAZ9-OsMYC2 interaction was decreased in the presence of OsNPR1 protein (Fig5c, Lines 323-326). From these results, is there a possibility that P2 enhanced the OsJAZ9-OsMYC2 interaction? To make it easier to understand, please add the explanation and discussion on the working model of the viral counter-defense mechanism, especially on the interaction between P2 and NPR1/MYC/JAZ/MED25 complex based on the evidence from this and previous articles such as the paper by Li et al (2021, PNAS) showing that P2 weakened the interaction between OsMYC3 and OsMED25.

Reviewer #3 (Remarks to the Author):

The authors addressed most of my previous comments/suggestions. They added several new experiments and elaborated on the sections that were unclear, all in a rather short turnaround, which is quite remarkable. A few comments I have left are below.

The conclusion that NPR1 levels are "significantly decreased" in the P2OX/NPR1OX cytosolic fractions is not convincing (Figure S4b, Author Response Fig. 9). Based on visuals (or pixel quantification) alone, the changes in NPR1 levels among the NPR1OX and P2OX/NPR1OX lines appear to be minimal, less so when compared/normalized to the levels of actin and histone protein markers that are also variable.

The authors refer to the immunoblot differences in the narrative at several places (e.g., lines 167, 198, 207, and others) as "significantly" different when no statistical significance tests were performed. Please correct the statements accordingly throughout the narrative or perform the appropriate statistical tests.

In the results/methods pertaining to the disease index scoring (Fig. 4b and 4f, and Author Response Fig. 11), the authors added more specifics that were previously unclear. However, again, they refer to this data as "statistical analysis", but no statistical tests are described or p-value cutoffs indicated in the figures or legend. Please correct the statements accordingly or perform the appropriate statistical tests.

Response to Reviewers' Comments

We are very grateful to the reviewers for their constructive and valuable suggestions on the manuscript. We have carefully taken these into consideration in preparing this revision and have performed the additional experiments suggested. We believe that the revised manuscript has addressed the reviewers' concerns and has resulted in a better paper. Comments received are shown in black below, with our response in red font. The following are our point-by-point answers to the reviewers' questions and comments.

Reviewer #1 (Remarks to the Author):

The authors have addresses all of my major and minor concerns. I am satisfied with the authors' answers and changes and have no more questions and comments.

Response: We thank the Reviewer#1 for the positive comments.

Reviewer #2 (Remarks to the Author):

I thank the authors for their careful revisions. The paper has been significantly improved. The authors have addressed my comments, but two points need to be considered.

Response: We thank reviewer #2 for the positive comments and have tried to incorporate further experiments in the revised manuscript as suggested, which have improved the manuscript significantly (see below).

1) (Regarding the first major points)

To draw the authors' conclusion on the OsNPR1 function in the transcriptional regulation of JA genes, further evidence is required on how OsNPR1 activates JA-regulated gene expression in concert with OsMYC2. To investigate whether

NPR1 targets MYC2 at actively transcribed JA-responsive genes. Are there any data on it? Is the OsNPR1/OsMYC2 complex actually recruited in the promoter region of JA-regulated genes such as *OsMADS1* and *OsNOMT* in vivo? does OsJAZs dissociate from the promoter region by OsNPR1?

Response: Thanks for this comment. In the revised manuscript, we have conducted chromatin immunoprecipitation ChIP-qPCR on various transgenic plants using OsNPR1-specific polyclonal antibodies to investigate how OsNPR1 activates OsMYC2-mediated JA-responsive genes *in vivo* (Fig. 5e-g, Line 340-358). Firstly, we performed ChIP-qPCR using OsMYC2-specific polyclonal antibodies in NIP or *Ri-m2m3* plants, in which the expression of OsMYC2 and OsMYC3 were significantly decreased, to prove OsMYC2 could bind to the promoters of the *OsMADS1* and *OsNOMT* genes *in vivo*. As showed in below Figure 1 (Supplementary Fig. 11a, b in revised manuscript), OsMYC2 could specifically bind to the G-box motif in the promoters of the *OsMADS1* and *OsNOMT* genes.

Figure 1. ChIP-qPCR assays indicated that OsMYC2 binds to the promoters of the *OsMADS1* and *OsNOMT* genes.

Then, we further performed ChIP-qPCR in NIP and *Ri-m2m3* mutant plants using OsNPR1-specific polyclonal antibodies (Fig. 5e, f, Line 348-355). Surprisingly, OsNPR1 specifically bound to the G-box motif but not to other regions in the promoters of *OsMADS1* and *OsNOMT* in wildtype NIP. The ability of OsNPR1 to bind to these promoters of *OsMADS1* and *OsNOMT* was significantly decreased in *Ri-m2m3* mutant plants (See Figure 2 below). These results indicated that OsNPR1 is actually recruited in the promoter region of JA-regulated genes by associating with OsMYC2.

Figure 2. ChIP-qPCR assays showed that the association of OsNPR1 with the promoters of the *OsMADS1* and *OsNOMT* genes depended on OsMYC2/3.

Thirdly, we performed ChIP-qPCR assays on OsNPR1-overexpressing (*OsNPR1-7#*) plants and its wildtype (NIP) plants. The results showed that the enrichments in the promoters of JA-responsive genes immunoprecipitated by OsNPR1 were increased in OsNPR1-overexpressing plants compared to NIP plants (see Figure 3 below and Fig. 5g in revised manuscript). These results further indicated that OsNPR1 induced JA-responsive genes by forming a complex with OsMYC2 at G-box motifs.

Figure 3. ChIP-qPCR assays showed that OsNPR1 binds to the G-box from *OsMADS1* and *OsNOMT* promoters.

In addition, we added a competitive Co-IP assay using *npr1-cas* mutant plants to address whether OsNPR1 dissociates the OsJAZ-OsMYC complex *in vivo* in revised manuscript (Supplementary Fig. 9d-f, Line 327-332). We found that OsJAZ9-OsMYC2 interaction was enhanced in the absence of OsNPR1 protein in rice. The results further suggested that OsNPR1 interrupted the OsJAZ9-OsMYC2/3 interaction (See Figure 4 below). Combined with the aboved ChIP-qPCR assays, we conclude that OsNPR1 dissociates the OsMYC2-OsJAZs interaction to activate the expression of JA-responsive genes.

Figure 4. OsNPR1 dissociated the OsJAZ9-OsMYC2 complex *in vivo*.

2) (Regarding the third major point)

The author mentioned in the rebuttal letter, “We have previously shown that P2 protein did not disturb the association of OsJAZ11 and OsMYC2/3, ...”. In this article, P2 accelerated the degradation of OsNPR1 (Fig2), and OsJAZ9-OsMYC2 interaction was decreased in the presence of OsNPR1 protein (Fig5c, Lines 323-326). From these results, is there a possibility that P2 enhanced the OsJAZ9-OsMYC2 interaction?

Response: To solve the reviewer’s concern, we have performed BiFC assays to make it clear whether P2 affected OsJAZ9-OsMYC2 interaction in the presence of OsNPR1. The results showed that the fluorescence formed by OsMYC2-cYFP and nYFP-OsJAZ9 was visibly increased when co-expressed with P2 and OsNPR1 compared to OsNPR1 alone. Although P2 protein did not disturb the association of OsJAZ9 and OsMYC2/3, our results suggested that P2 could indirectly stable OsJAZ9-OsMYC2 complex by relieving the interruption of OsNPR1 on JAZ9-MYC2 complex (see Figure 5 below).

Figure 5. P2 could relieve the inhibition of OsNPR1 on JAZ9-MYC2 complex.

To make it easier to understand, please add the explanation and discussion on the working model of the viral counter-defense mechanism, especially on the

interaction between P2 and NPR1/MYC/JAZ/MED25 complex based on the evidence from this and previous articles such as the paper by Li et al (2021, PNAS) showing that P2 weakened the interaction between OsMYC3 and OsMED25.

Response: We have added more detailed descriptions on discussion and Fig. 8 legend, and revised the model in the revised manuscript (Line 595-602 and line 1415-1422).

Reviewer #3 (Remarks to the Author):

The authors addressed most of my previous comments/suggestions. They added several new experiments and elaborated on the sections that were unclear, all in a rather short turnaround, which is quite remarkable. A few comments I have left are below.

Response: We thank the Reviewer#3 for the positive comments.

The conclusion that NPR1 levels are “significantly decreased” in the P2-OX/NPR1-OX cytosolic fractions is not convincing (Figure S4b, Author Response Fig. 9). Based on visuals (or pixel quantification) alone, the changes in NPR1 levels among the NPR1OX and P2OX/NPR1OX lines appear to be minimal, less so when compared/normalized to the levels of actin and histone protein markers that are also variable.

Response: To address the reviewers' concern, we re-performed the western blotting analysis using the nuclear and cytoplasmic fractions of *OsNPR1-OX* and *OsNPR1-OX/P2-OX* leaves, under the condition of consistent protein levels of actin and histone (Supplementary Fig. 4b). The results showed that P2 visibly reduced the level of OsNPR1 in the cytoplasm, while only a small reduction was found in the nucleus (see Figure 6 below).

Figure 6. Nuclear-cytoplasmic fractionation analysis of OsNPR1 accumulation in *OsNPR1-OX* and *OsNPR1-OX/P2-OX* plants.

The authors refer to the immunoblot differences in the narrative at several places (e.g., lines 167, 198, 207, and others) as “significantly” different when no statistical significance tests were performed. Please correct the statements accordingly throughout the narrative or perform the appropriate statistical tests.

Response: We are very grateful to Reviewer #3 and have corrected the statements carefully in our revised manuscript as suggested (Line 167, 198, 207 and 570).

In the results/methods pertaining to the disease index scoring (Fig. 4b and 4f, and Author Response Fig. 11), the authors added more specifics that were previously unclear. However, again, they refer to this data as “statistical analysis”, but no statistical tests are described or p-value cutoffs indicated in the figures or legend. Please correct the statements accordingly or perform the appropriate statistical tests.

Response: We thank the reviewer for this comment. We have added data listed in Supplementary Table 2 and carried out statistical tests using ORIGIN

8.0 software in the revised manuscript (see Figure 7 below).

Figure 7. A statistical analysis of the disease index in *OsNPR1-OX* and *npr1-cas* mutant plants.

Reviewer #2 (Remarks to the Author):

I thank the authors for their revisions.
The authors have addressed all of my concerns.
I have no more questions and comments.

Reviewer #3 (Remarks to the Author):

Authors have addressed my previous comments, and I have no additional comments.